# CAESAR: An Embodied Simulator for Generating Multimodal Referring Expression Datasets

**Md Mofijul Islam, Reza Manuel Mirzaiee, Alexi Gladstone, Haley N. Green, Tariq Iqbal**
School of Engineering and Applied Science, University of Virginia, Charlottesville, USA
{mi8uu, rmm3ya, abg4br, hng9vf, tiqbal}@virginia.edu

## Abstract

Humans naturally use verbal utterances and nonverbal gestures to refer to various objects (known as *referring expressions*) in different interactional scenarios. As collecting real human interaction datasets are costly and laborious, synthetic datasets are often used to train models to unambiguously detect relationships among objects. However, existing synthetic data generation tools that provide referring expressions generally neglect nonverbal gestures. Additionally, while a few small-scale datasets contain multimodal cues (verbal and nonverbal), these datasets only capture the nonverbal gestures from an exo-centric perspective (observer). As models can use complementary information from multimodal cues to recognize referring expressions, generating multimodal data from multiple views can help to develop robust models. To address these critical issues, in this paper, we present a novel embodied simulator, CAESAR, to generate multimodal referring expressions containing both verbal utterances and nonverbal cues captured from multiple views. Using our simulator, we have generated two large-scale embodied referring expression datasets, which we have released publicly [1]. We have conducted experimental analyses on embodied spatial relation grounding using various state-of-the-art baseline models. Our experimental results suggest that visual perspective affects the models' performance; and that nonverbal cues improve spatial relation grounding accuracy. Finally, we will release the simulator publicly to allow researchers to generate new embodied interaction datasets.

## 1 Introduction

Natural communication forms of humans are inherently multimodal with verbal and nonverbal (gestures and gaze) signals [1–5]. People use multimodal cues with their interaction partners for the joint focus of attention on salient objects and events, specifically when they share a physical space in an environment [3–9]. As humans use multimodal communication forms for interactions, we need AI-driven agents interacting with us to understand multimodal referring expressions to generate seamless interactions [3, 10–12].

Comprehending referring expressions has been generally studied in the form of the *spatial relation grounding task* [13–26]. This task involves identifying whether the verbal utterance of the spatial relationships between objects holds in the visual scene [15, 16]. However, the exclusion of nonverbal signals in the model makes the problem different from how people interact naturally in shared physical spaces, as people start to use multimodal signals very early in their developmental phase [1, 2, 5, 27–32]. To address this gap, in this work, we have designed an *embodied spatial relation grounding task*, which involves identifying whether a person is verbally and nonverbally (pointing gesture and gaze) referring to the same objects in the visual scene. This task can help develop learning frameworks to understand multimodal referring expressions in embodied settings.

---

[1]CAESAR Datasets: `caesar-simulator.github.io`

36th Conference on Neural Information Processing Systems (NeurIPS 2022) Track on Datasets and Benchmarks.

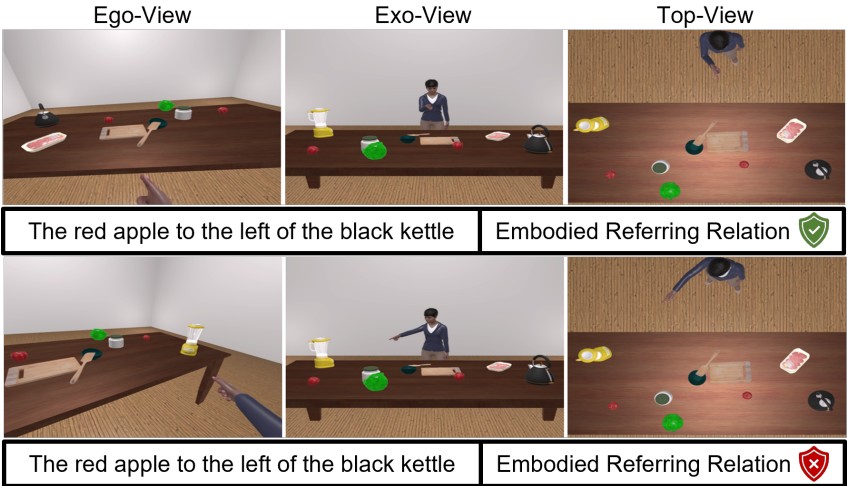

Figure 1: Embodied referring expressions generated using, CAESAR, with verbal and nonverbal modalities from multiple views. Top: verbal utterance and nonverbal gestures both referring to the same object (i.e., Apple). Bottom: verbal utterance refers to the Apple; however, the nonverbal gestures refer to the Blender.

A few datasets have been developed to capture embodied multimodal referring expressions, which involve referring an object using verbal utterances and nonverbal cues (pointing gesture and gaze) [3, 33]. However, these datasets have several crucial limitations. The primary limitation of existing datasets is that the nonverbal interactions are captured solely from an exocentric perspective (*exo, ego, and top* view denotes perspectives from an actor, the observer, and overhead, respectively (Fig. 1)). As comprehending embodied referring expression requires perspective-taking, which is the awareness of the actor's and observer's point of view in shared interactions, the lack of perspective-awareness in these datasets can degrade the model's performance. Additionally, multiple views can help identify the referred object, which may be partially occluded from one view but visible from another. Moreover, in human-human interactions, learning perspective is used innately to attend to salient parts of interactions. Let's assume an actor is requesting an observer verbally to "pick up the left apple" (Fig. 1). This verbal expression can be interpreted differently from different perspectives, where the "left apple" from the exo view can be interpreted as the "right apple" from the ego view. Learning where the actor is looking and pointing can help identify the appropriate object in these scenarios. These data samples with multiple views enable the model to learn perspective-taking to ensure seamless and natural interactions in embodied settings.

Additionally, contrastive verbal and nonverbal expressions are common in many real-world settings. For example, humans often mistakenly describe one object while pointing to another object. People are adept at identifying these scenarios and involve themselves in a conversation to complete the communication. Similarly, an intelligent AI-agent should identify inconsistent interactions from multimodal referring expressions. However, existing datasets only contain congruent and complete verbal and nonverbal interaction signals. Therefore, to train a robust model, we need a dataset with contrastive data samples, enabling the agent to request additional information from human partners in cases of incongruent signals.

To address the shortcomings of the existing datasets, we have developed a novel embodied simulator, CAESAR, to generate large-scale datasets of referring expressions. To the best of our knowledge, CAESAR is the first simulator to generate multimodal referring expressions with verbal utterances and nonverbal gestures in a virtual environment. CAESAR has three novel aspects which differentiate it from other synthetic data generation systems (e.g., CLEVR [34] and Kubric [35]). First, CAESAR simulates scenarios in which verbal utterances and nonverbal cues (pointing gesture and gaze) refer to objects in an embodied setting (Fig. 1). We have collected real human pointing gesture data using an OptiTrack motion capture system [36] and emulated the same behaviors in CAESAR by incorporating a new stochastic deictic gesture generation approach. Second, CAESAR renders multiple views from different perspectives, such as ego-, exo-, and top-view, that can aid in training models to learn different perspectives for comprehending multimodal referring expressions. Third, taking inspiration from previous work [15], we have designed a module in CAESAR to generate contrastive samples, where the virtual human is pointing to an object while verbally describing a different object.

Table 1: Comparison of the datasets of referring expression comprehension. V, NV, E, C, and A denote verbal, nonverbal, embodied, contrastive samples, and ambiguous samples, respectively. *Average number of words.

| Datasets | V | NV | E | Views | | | C | A | No. of Images | No. of Samples | Object Categories | Avg. Words* |
|---|---|---|---|---|---|---|---|---|---|---|---|---|
| | | | | Exo | Ego | Top | | | | | | |
| PointAt [37] | ✗ | ✓ | ✓ | ✓ | ✗ | ✗ | ✗ | ✗ | 220 | 220 | 28 | - |
| ReferAt [33] | ✓ | ✓ | ✓ | ✓ | ✗ | ✗ | ✗ | ✗ | 242 | 242 | 28 | - |
| IPO [38] | ✗ | ✓ | ✓ | ✓ | ✗ | ✗ | ✗ | ✗ | 278 | 278 | 10 | - |
| IMHF [39] | ✗ | ✓ | ✓ | ✓ | ✗ | ✗ | ✗ | ✗ | 1716 | 1716 | 28 | - |
| RefIt [40] | ✓ | ✗ | ✗ | ✓ | ✗ | ✗ | ✗ | ✗ | 19,894 | 130,525 | 238 | 3.61 |
| RefCOCO [41] | ✓ | ✗ | ✗ | ✓ | ✗ | ✗ | ✗ | ✗ | 19,994 | 142,209 | 80 | 3.61 |
| RefCOCO+ [41] | ✓ | ✗ | ✗ | ✓ | ✗ | ✗ | ✗ | ✗ | 19,992 | 141,564 | 80 | 3.53 |
| RefCOCOg [42] | ✓ | ✗ | ✗ | ✓ | ✗ | ✗ | ✗ | ✗ | 26,711 | 104,560 | 80 | 8.43 |
| Flickr30k [43] | ✓ | ✗ | ✗ | ✓ | ✗ | ✗ | ✗ | ✗ | 31,783 | 158,280 | 44,518 | - |
| GuessWhat? [44] | ✓ | ✗ | ✗ | ✓ | ✗ | ✗ | ✗ | ✗ | 66,537 | 155,280 | - | - |
| Cops-Ref [45] | ✓ | ✗ | ✗ | ✓ | ✗ | ✗ | ✗ | ✗ | 75,299 | 148,712 | 508 | 14.40 |
| CLEVR-Ref+ [46] | ✓ | ✗ | ✗ | ✓ | ✗ | ✗ | ✗ | ✗ | 99,992 | 998,743 | 3 | 22.40 |
| YouRefIt [3] | ✓ | ✓ | ✓ | ✓ | ✗ | ✗ | ✗ | ✗ | 497,348 | 4,195 | 395 | 3.73 |
| CAESAR-L | ✓ | ✓ | ✓ | ✓ | ✓ | ✓ | ✓ | ✓ | 11,617,626 | 124,412 | 61 | 5.56 |
| CAESAR-XL | ✓ | ✓ | ✓ | ✓ | ✓ | ✓ | ✓ | ✓ | 841,620 | 1,367,305 | 80 | 5.32 |

One of the primary goals of developing CAESAR is to democratize the data generation process so that researchers without simulator development experience can have complete control of generating a diverse dataset to train and evaluate a learning model. Similar to existing data generation systems, the development of our simulator requires extensive knowledge of motion planning and game engine. Thus, to make it accessible to everyone, we have developed a tool that enables researchers to generate diverse samples without any simulator development experience. Using this tool, we have developed two large-scale datasets, CAESAR-XL and CAESAR-L, for understanding multimodal referring expression in an embodied virtual environment. A comparison of our developed datasets and other existing datasets for referring expression understanding is listed in Table 1.

Although several state-of-the-art visual-language models have been proposed for different tasks, such as spatial relation recognition [14–16], referring expression comprehension and visual question answering [21, 20, 22], these approaches use nonverbal embodied interactions from only an exocentric perspective. Thus, we have adopted state-of-the-art models and benchmarked on our datasets for grounding embodied spatial relations using multiple views and multimodal data. Our experimental results suggest that these models' performance varies with perspective, and nonverbal cues can improve it. Moreover, the results also indicate that we need to develop models that extract salient nonverbal cues and effectively fuse verbal utterances for robust performance.

The key contributions of this work are listed below:

- We have developed a novel embodied simulator, CAESAR, to generate referring expressions with verbal uttrances and nonverbal gestures captured from multiple perspectives.

- We have generated two large-scale and one small datasets of multimodal referring expressions in an embodied setting using CAESAR.

- We have benchmarked various models on our dataset. The results suggest these models cannot effectively learn perspective-taking, which opens new research directions to develop robust models for embodied referring expression comprehension.

- CAESAR allows researchers to tune the simulator's parameters without any development experiences to generate customized samples for training and diagnosing their models.

## 2   Related Work

**Spatial Relation Grounding Datasets:** Several synthetic datasets of various images of objects are generated for spatial relations grounding tasks using game engines, such as Unity [15, 16, 46–48]. For example, Goyal et al. [15] uses Blender to generate synthetic dataset, Rel3D, for spatial relationship recognition. However, in this dataset, the visual scene only contains two objects which simplifies the task. Lee et al. [16] addresses issues of Rel3D by generating multiple objects in the scene. Unlike these synthetic datasets, other datasets use real-world images [40–42]. For example, SpatialSense

[14] uses real-world images for spatial relationship detection. In these datasets, verbal referring expressions are generated using either template-based methods [49, 50, 46] or human annotators [41, 40, 49, 14, 51]. For example, the ReferIt3D dataset [49] uses a compositional template (<target> <spatial-relation> <reference>) to generate verbal utterances. However, one of the limitations of these datasets is the absence of the nonverbal cues (pointing gestures and gaze).

**Datasets Generator:** Existing synthetic data generation tools [15, 16, 46, 34] work adequately for generating referring expressions in non-embodied settings. For example, the GRiD-3D dataset [16] uses Blender [52] to generate referring expressions for relation grounding, object identification, and visual question answering. However, these tools were not designed to generate nonverbal gestures in embodied settings. Moreover, it is non-trivial to extend the existing simulators [15, 16, 46, 34] and procedurally generate non-verbal gestures. In our previous work [53], we found that a template-based approach produces pointing gestures that deviate from realistic gesture.

**Embodied Spatial Relation Grounding Datasets:** A few datasets have been developed for embodied referring expression comprehension [3, 33, 37–39]. For example, the YouRefIt [3] dataset was developed in a real-world setting, which has several advantages over synthetic data. However, this dataset is limited in sample size and lack detailed annotations, which only contains 4,195 unique visual scenes. Moreover, the nonverbal interactions in the existing datasets have been captured only from the exocentric view, which may limit the model's ability to learn multiple perspectives. Although prior works have shown the importance of contrastive data samples to train models [15], the existing datasets of embodied referring expressions do not include contrastive data samples. Furthermore, the existing datasets do not include ambiguous expressions, which can be used to develop conversational embodied agents to ensure seamless human-AI interactions. Additionally, the existing datasets do not explicitly consider generating samples with the occluded objects from a particular view, which can help diagnose the model's robustness to ground embodied spatial relations. A comparison of various referring expression datasets has been presented in Table 1.

## 3   CAESAR: An Embodied Simulator

In this section, we present CAESAR, an embodied simulator capable of automatically generating multimodal referring expressions with verbal utterances and nonverbal cues (pointing gesture and gaze) to refer to an object. Generated embodied referring expressions are depicted in Fig. 1.

For CAESAR we have created an environment where an embodied agent (an avatar) refers to various objects distributed on a table top through nonverbal gestures and verbal utterances by exploiting spatial relation with other objects in the scene. CAESAR generates the environment by dynamically loading various avatars, objects, walls, and tables.

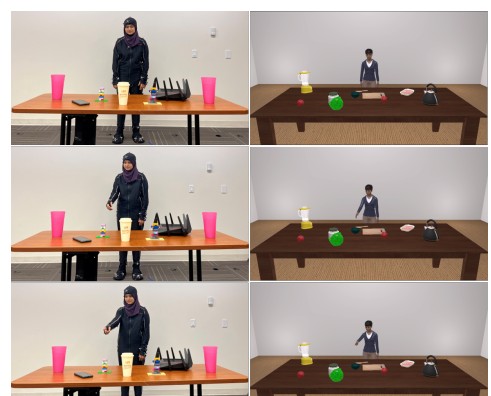

Figure 2: Comparison of real (left) and synthetic motion generated from CAESAR (right). We used real human motion using an OptiTrack motion capture system to synthesize gestures in our simulator.

### 3.1   Observer-Aware Object Generator

To ensure plenty of variation across data samples while limiting clutter, CAESAR randomly spawns between four to ten objects from our pre-populated object library. Among these spawned objects, CAESAR randomly chooses one object as the referred object, which will later be described through nonverbal cues and verbal utterances. We apply some constraints to an object to be declared properly generated. First, objects can only occur in a scene at most three times. Second, objects must be partially visible from both the ego and exo views. To promote object diversity, CAESAR varies spawned objects in rotation, color, size, and position. CAESAR does not vary some object colors, such as oranges, to ensure proper object appearance.

### 3.2   Embodied Referring Expression Generator

CAESAR generates both verbal and nonverbal referring expressions for each embodied interaction. To vary nonverbal expressions, different cues are used interchangeably by the human avatar to refer

Table 2: Verbal referring expression generation templates. Here, <Obj>: Referred object name, <Obj-1>: Reference object name, <Obj-n Prop.>: Color or Size of object $n$, <SR>: Spatial relation. Note that spatial relations/locations are relative to either the observer (exo view) or embodied agent (ego view).

| No. | Template of Verbal Referring Expression | Example Instance |
|-----|------------------------------------------|------------------|
| 1 | <Obj> | The apple |
| 2 | <Obj Prop.><Obj> | The red apple |
| 3 | <Spatial Location><Obj Prop.><Obj> | The center small apple |
| 4 | <Obj><SR><Obj-1 Prop.><Obj-1> | The apple to the left of the big cutting board |
| 5 | <Obj Prop.><Obj><SR><Obj-1 Prop.><Obj-1> | The small apple next to the brown cutting board |

to objects. Nonverbal data consists of procedurally generated pointing gestures and a gaze that refers to an object. To achieve this, we dynamically calculate musculo-skeletal motion for eight different avatars [53, 54]. Additionally, verbal expressions are generated randomly from a set of templates.

**Pointing Gesture Synthesis:** One of the primary goals of the CAESAR simulator is to generate realistic pointing gestures that match the amount of variability in real human gestures. To accomplish this, we have researched prior works for procedural pointing gesture synthesis and developed a novel synthetic pointing gesture synthesis algorithm. Past algorithms generally fall under three categories: data-driven algorithms that use motion-capture data to fit constraints [55], physics-driven algorithms that build motion using a musculo-skeletal simulation [56], and hybrid algorithms combine aspects of the prior two. Our method of synthesizing motion [53] is a hybrid algorithm that synthesizes a motion path for pointing gestures based on motion-capture data (using similar timing and arc-like motion). We also determine joint rotations based on inverse kinematics to fit this motion path, and subsequently applies a physical simulation to account for gravity, momentum, and self-collision. We have used real human motion data using an OptiTrack motion capture system to synthesize human pointing gestures in CAESAR (Fig. 2).

Our gesture generation algorithm has five phases: rest, preparation, stroke, hold, and retraction. The rest phase consists of a static idle animation; the subsequent phases are layered onto this to simulate the micro-movements that occur while muscles are under tension. For the preparation, stroke, and hold phases of the gesture, the motion path is determined by constructing a Catmull-Rom curve [57] through a set of three points: one at the rest-coordinates of the pointing hand, one at the peak-coordinates, where the hand is most extended in a pointing gesture, and a third point at the midpoint of the previous points, with a varying displacement as to randomly alter the shape of the motion. This curve is then converted into a Bezier Curve [58]. To allow human hands to travel along paths we used 3D Bezier Curves [59]. We also added a Cubic Easing function along the path and a basic physical particle simulation to the path-following object as a basis for our Two-Joint Inverse Kinematics (IK) target when creating the arm animation. To implement the IK and physical simulation, we used the Unity Animation Rigging [60] and Dynamic Bone [61] packages. The retraction phase of the gesture is implemented by easing off the IK constraint's strength, allowing for gravity to swing the limb back into its rest position, directly under the shoulder joint.

**Gaze Synthesis:** Our avatars' head and body orientation is calculated through a set of IK targets, also using the Unity Animation Rigging package [60]. They are layered on top of each other: first, the body is applied, and then the head is applied. The IK target weights are eased in according to a timing parameter shared between the pointing and gaze systems. This constraint ensures that when both gaze and pointing gestures are generated, the avatar will look towards the target and change their body's orientation before pointing.

**Verbal Referring Expression Generation:** Taking inspiration from previous works [49, 50, 46], we have developed five compositional templates to generate verbal referring expression, presented in Table 2. In these templates, the target object is referred to by verbal and nonverbal cues, a reference object is used to add context to the target object's location, and the object's properties, such as color, size, and spatial location (left, right, corner), are varied. For example, using Template-5, we can generate the verbal message "the red apple to the right of the black kettle", depicted in Fig. 1. We also varied the spatial relation/location of the target object by referring to it from either the observer's or the actor's perspectives, resulting in twelve verbal expressions formulated from the five templates. Please see the supplementary for further details.

### 3.3 Rendering Nonverbal Referring Expressions from Multiple Views

CAESAR generates nonverbal referring expressions in three scenarios - a person gazing at an object, a person pointing to an object, and a person gazing and pointing at an object. There is another setting where no human avatar is rendered and the scene only contains objects, named *no human* scenario. Nonverbal cues in three of the scenarios are captured from three camera views: ego, exo and top. We have also generated skeletal poses of a simulated human avatar using the Vectrosity package [62].

### 3.4 Contrastive Sample Generator

CAESAR generates contrastive embodied referring expressions where the given embodied referring expressions are insufficient to successfully ground an object. As described in Section 3.3, there are four different scenarios our simulator generates; CAESAR generates contrastive embodied referring expressions for all four of these different scenarios. In the situations of only gaze, only pointing gesture, and both gaze and pointing gesture, the human avatar points, gazes or both at an object that it is not verbally describing. This is made apparent in Fig. 1, where the humanoid verbally and nonverbally describing two different objects results in contrastive expressions. For the scenarios with *no human* avatar, CAESAR generates a verbal expression that describes an object not in the scene. These contrastive data samples can help to train models to ground embodied spatial relations. While generating these contrastive scenarios, we apply several constraints to ensure non-ambiguity in whether a scenario is contrastive or not. For example, a chosen object's proximity to different copies of that object is checked to ensure that a referred object can be distinguished from other referred objects. Additionally, the contrastive object selected for nonverbal expression is checked for being adequately spaced from the chosen object and of a different category. These constraints ensure the person's gaze or gesture is sufficiently different to make the sample contrastive.

### 3.5 Data Annotation

CAESAR generates detailed annotations of each data sample, including bounding box coordinates for all the generated objects from all three views, object attributes (color, size, absolute location), and their relative locations from the actor (ego view) and the observer (exo view). We found that using Unity's object mesh renderer for bounding box calculations provided large overestimates, so we calculate the position ($x$ and $y$ coordinates) of each vertex of an object relative to each camera, which leads to accurate bounding box annotations. Additionally, as the ego view camera constantly changes position and rotation (the other cameras remain static), we dynamically calculate bounding boxes during videos for the ego camera to effectively track where each object is relative to the moving camera. Moreover, CAESAR annotates each verbal referring expression according to object attributes and spatial relations. It also records environmental parameters, such as lighting conditions (number of lights, position, intensity) and background color. Please check the supplementary materials for more details of data annotation as well as dynamic bounding box calculation.

### 3.6 Configurable Data Generation Interface

One of the challenges of many simulators that generate datasets is a lack of configurability or a requirement of extensive development experiences using certain libraries. We have developed a tool to configure CAESAR without programming or game engine experience through simply clicking buttons in the Unity inspector window to toggle features. This tool uses serialized fields inside our main manager, directly allowing users to configure different features that will be used internally. These configurable features include the ability to specify whether video should be recorded, activate different cameras (i.e, the skeletal camera), designate the number of scenes to generate in parallel, as well as many other features described in more detail in the supplementary document.

## 4 Dataset Analyses

Using CAESAR, we have generated two datasets. The first dataset, CAESAR-L, consists of $124,412$ data samples created from $11,617,626$ images at a resolution of $480 \times 320$ pixels. These data samples are divided into train, validation, and test data splits with $74,760$, $24,779$, and $24,873$ data samples, respectively. The second dataset, CAESAR-XL, consists of $1,367,305$ data samples, which were created from $841,620$ images by varying verbal expressions in the five different settings described in

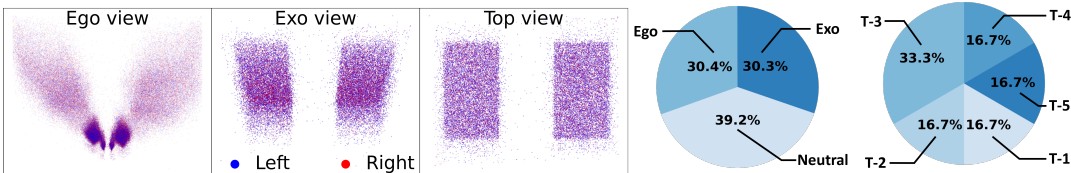

(a) Referred object locations in different views.  (b) Distribution of perspectives. (c) Distribution of templates.

Figure 3: Analysis of CAESAR-XL dataset. (a) CAESAR-XL has little to no bias as spatial-visual cues of object locations are less separable for a given *left* and *right* spatial location in verbal utterances. Note that the color being purple is a result of overlapping left and right points. (b) CAESAR-XL contains referring expressions from all the perspectives: ego (actor), exo (observer), and neutral (expressions that do not depend on perspective-taking). (c) CAESAR-XL generates verbal utterances using the templates described in Table 2. ($T - n$ denotes the $n - th$ template).

Section 3.3. These data samples are divided into train, validation, and test data splits with $1,123,886$, $122,157$, and $121,262$ data samples, respectively. These images were rendered with a resolution of $720 \times 480$ pixels using an object pool of size 80. The lower sample to image ratio in CAESAR-L dataset when compared to CAESAR-XL can be explained by the CAESAR-L dataset containing images, videos (rendered at 15 fps), and a skeletal pose. Table 1 shows an in-depth comparison between CAESAR-L, CAESAR-XL, and other similar datasets from the literature.

Similar to previous works [14, 15], one of the primary goals of CAESAR is to reduce the spatial location bias in generated data. For example, if the terms *"on the left"* and *"on the right"* always refer to objects located on the left or right side of the scene from the actor's perspective, models will exploit this bias to ground spatial relations. If actors give utterances such as *"on the left"* and *"on the right"* but from the view of the observer, instead of the view of the actor, these models will not be able to complete the perspective taking necessary to successfully ground these utterances. To address the issue, we randomly select verbal expressions from either the ego (actor) or exo (observer) perspective. We visualize the referred object location in each view (ego, exo, and top) for left and right spatial locations (Fig. 3(a)). These visualizations suggest that referred objects of these two locations are spread across both sides of the three views, meaning the object locations are not identifiable through solely verbal cues. This analysis indicates that our datasets are not biased in generating spatial locations, and thus, can force models to utilize nonverbal cues to succeed in embodied spatial relation grounding by recognizing which perspective given utterances come from.

As shown in Fig. 3(b), our datasets contain verbal utterances from multiple perspectives (ego, exo, neutral). Neutral utterances do not depend perspective to ground relations. For example, the term "the red apple" does not contain any spatial relation/location terms and thus does not require perspective to ground. The relatively equal distribution of multiple perspectives in our datasets promote the ability for models to learn perspective-taking in embodied settings. Fig. 3(c) shows the distribution of each verbal expression template presented in Table 2, where template three (the template involving spatial location, an object property, and then an object) was about twice as common as all other templates. This was done to ensure spatial relations and spatial locations were used at the same frequency (as templates four and five both use spatial relations). Thus, our dataset is not biased towards verbal expressions. Please check the supplementary for further analyses of our datasets.

## 5  Embodied Relation Grounding Models

Existing models use verbal utterance and an exocentric view to recognize spatial relations. However, our dataset contains both verbal and nonverbal modalities captured from three views. Thus, we have adopted visual-language models to develop three representations learning models for the embodied spatial relation grounding task: a CLIP Model [63], a Dual-Encoder (ViT [64] + BERT [65]) model and a Late Fusion (ResNet [66] + BERT [65]) model (Fig. 4).

**CLIP-based Model:**  CLIP model excels at aligning visual and language modalities [63]. Thus, we use the CLIP model to detect whether nonverbal cues and verbal utterances of an embodied expression refer to the same object. However, CLIP can take an image-text pair and produce verbal and visual representations. For this reason, we pair the verbal expression, $T$, to each of the views of the nonverbal expression (Ego ($V_{ego}$), Exo ($V_{exo}$), and Top ($V_{top}$)) and pass each modality pair

to CLIP models: $E_i^v, E_i^t = CLIP(V_i, T), i \in (ego, exo, top)$. Here, $E_i^v \in \mathbb{R}^{B \times S}$ and $E_i^t \in \mathbb{R}^{B \times S}$ are the visual and verbal embeddings from CLIP models, respectively. ($B$ is the batch size and $S$ is the embedding dimension)

**Visual-Language Transformer Models:** We have extended two visual-language transformer models, ViLT [67] and VisualBERT [21], for grounding embodied relations. As these models were designed to produce representations from a single visual scene and a verbal utterance, we extend these models to extract visual-language representations from multiple visual scenes. First, we extract visual representations from multiple visual scenes using ResNet-50. Finally, we pass these visual tokens and tokenized verbal utterance to these visual-language transformer models. These models process these visual and verbal tokens using a single transformer model to extract combined visual-language representation. This representation is then used for grounding embodied relations.

**Dual-Encoder Model:** Like CLIP models, we pass each pair of visual and verbal modalities to Dual-Encoder models to extract verbal and non-verbal representations. We use ViT and BERT in Dual-Encoder models to encode visual and verbal modalities, respectively. Both of these encoders (ViT and BERT) use a Transformer to extract representations.

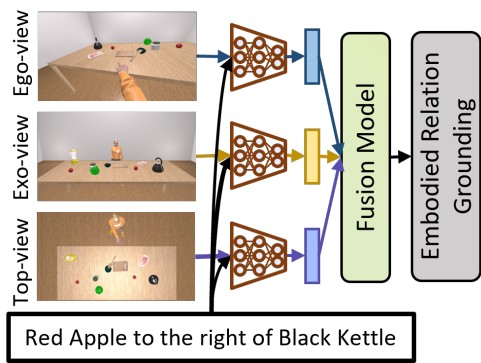

**Late Fusion Model:** In the Late Fusion model, all the visual and verbal modalities are encoded independently using ResNet-50 [66] and BERT [65] models. We projected the extracted verbal and visual modalities representations to a fixed-sized embedding. Although Dual-Encoder and Late-Fusion models have similar architectures, as both of these models use separate encoders for visual and verbal modalities, there are two main differences. First, Dual-Encoder models use Transformers to design both visual and verbal encoders. This contrasts from how Late-Fusion models use different architectures for these two types of modalities, such as ResNet for the visual encoder and BERT for the verbal encoder. Second, in the Late-Fusion model, we first extract visual representations for all three views and fuse the visual and verbal representations using a Transformer-style multi-head attention approach [68]. This differs from the Dual-Encoder model, where we pair the verbal utterances with each view and pass each visual-verbal pair through the model to extract pairwise representations, which are concatenated to produce multimodal representations.

Figure 4: Embodied relation grounding model. Data from each pair of verbal and visual modalities is passed through a shared visual language models to extract representations, which are then fused for embodied spatial relation grounding.

**Multimodal Fusion:** We fused the extracted verbal and visual representations from the above-mentioned models to produced multimodal representations, which are used to detect embodied spatial relation. We used four fusion approaches: SUM, CONCAT, Self-Attention, and Cross-Attention. The first two fusion approaches summed and concatenated the verbal and visual representations. The self-Attention approach is similar to the Transformer-style self-attention [68] which attends each of the verbal and visual representations and sums the attended representations. We have also employed a Cross-Attention approach, which is similar to the co-attention approach from ViLBERT [20]. Cross-Attention is essentially a query-key-value style attention approach, where verbal embeddings are used as query and visual embeddings are used as key and values. Finally, the fused embedding is passed through a multilayer perceptron to detect embodied spatial relations.

**Model Training Environment Setup:** We projected the visual and verbal embeddings from CLIP, Dual-Encoder, Late Fusion models to 512, 768, and 768 sized embeddings, respectively. We fused all the embeddings from multiple views and pass them through a multilayer perceptron to classify whether verbal and nonverbal expressions refer to the same object. We trained the model using Cross-entropy loss on the CAESAR-XL dataset for four epochs. To train our model we used PyTorch-Lightning [69] and HuggingFace [70] to implement models and used the Adam optimizer with weight decay regularization [71]. An initial learning rate set to $3e^{-4}$ to train the models. We trained all the models in a distributed GPU cluster environment, where each node contains 4-8 GPUs. Please check the supplementary materials for detailed training setup.

# 6 Results and Discussion

We evaluated several models on our dataset CAESAR-XL by varying modalities, perspectives, and fusion methods. To evaluate the impact of modalities (verbal and nonverbal), we used all the views (ego, exo, and top) and employed the CONCAT fusion method. Moreover, we used the CONCAT fusion method to evaluate the impact of various perspectives using verbal, gaze, and pointing gesture data. Additionally, we used the verbal utterances from ego, exo, and neutral perspectives to evaluate whether the baseline models can effectively learn perspective to ground embodied referring relation. Finally, to evaluate the impact of the fusion methods, we used all the views with verbal, gaze, and pointing gesture data. The results are presented in Table 3.

**Impact of modalities:** The results in Table 3(a) suggest that verbal models without any nonverbal signals (e.g., BERT [65]) can not perform better than random guessing at the relation grounding task. The reasoning behind this performance is that we generated contrastive nonverbal data samples for the same verbal utterance. Additionally, incorporating nonverbal modalities (gaze and pointing gesture) improve embodied spatial relation grounding accuracy. For example, incorporating only gaze cues improved the model's performance compared to models that use verbal and visual modalities only. This performance improvement indicates the necessity of nonverbal modalities to ground embodied spatial relations. However, the performance degrades when models use both gaze and gesture cues, when compared to models using only gaze or gesture. As the evaluated baseline models encode the visual modalities to extract combined representation for pointing gestures and gaze feature representations, these models may not disentangle pointing gesture and gaze representation to comprehend referring cues accurately. Thus, we must carefully design the model architecture and training procedure to extract cues from nonverbal modalities and effectively fuse these representations to verbal modality to accurately recognize embodied spatial relation.

**Impact of multiple perspectives:** The results in Table 3(b) suggest that the models' performance varies with the perspectives. For example,

Table 3: Embodied spatial relation grounding accuracy of baseline models. The results suggest that nonverbal cues increase embodied spatial relation grounding accuracy. However, the model's performance depends on how nonverbal interactions are captured and how representations from multiple views and modalities are fused. (V: Verbal, NH: Visual without Human, G: Gaze, P: Pointing Gesture, SA: Self-Attention, CA: Cross-Attention, LF: Late Fusion, DE: Dual-Encoder).

| (a) Impact of Modalities | | | | |
|---|---|---|---|---|
| Model | V | V+NH | V+G | V+P | V+G+P |
| BERT | 50.00 | - | - | - | - |
| LF | - | 78.44 | 81.51 | 81.18 | 76.00 |
| DE | - | 62.55 | 72.26 | 63.09 | 74.87 |
| CLIP | - | 64.78 | 83.01 | 77.31 | 75.21 |
| ViLT | - | 64.86 | 82.54 | 84.43 | 79.90 |
| VisualBERT | - | 68.07 | 80.07 | 77.94 | 75.61 |

| (b) Impact of Multi-Perspectives | | | |
|---|---|---|---|
| Model | Ego | Exo | Top | All |
| LF | 60.72 | 76.51 | 88.37 | 76.00 |
| DE | 59.75 | 68.84 | 71.04 | 74.87 |
| CLIP | 59.12 | 78.97 | 78.86 | 75.21 |
| ViLT | 85.43 | 62.47 | 52.12 | 79.90 |
| VisualBERT | 57.75 | 70.16 | 66.32 | 75.61 |

| (c) Impact of Fusion Methods | | | |
|---|---|---|---|
| Model | SUM | CONCAT | SA | CA |
| LF | 69.20 | 76.00 | 69.04 | 51.81 |
| DE | 77.02 | 74.87 | 72.50 | 74.89 |
| CLIP | 82.85 | 75.21 | 80.63 | 75.51 |

the top view improves the performance of Late Fusion and Dual Encoder models compared to models using the exo view. These findings underscore that the exo view is not always the optimal perspective for ensuring robust performance. Additionally, although the performance of the single view-based models fluctuates, incorporating multiple perspectives helps achieve consistent performance across the models. In our experiments, we found that multiview models cannot outperform single view-based models, unlike findings from previous works [72–79]. The reasoning behind this performance degradation is that nonverbal cues can be interpreted differently from multiple views. For example, a person pointing and verbally describing an object as the "left apple" can be visually interpreted by an observer as the "right apple". Thus, extracting synchronized cues across modalities is essential to validating an embodied spatial relation. Moreover, the evaluated baseline models do not explicitly learn to ground perspective to comprehend referring expressions. As the referring expressions in our datasets are generated from multiple perspectives, our datasets can be used to diagnose whether a model can effectively learn perspective-taking to comprehend embodied referring expressions.

**Impact of multimodal fusion:** The results in Table 3(c) suggest that simple SUM and CONCAT fusion approaches performed better than more complex attention-based fusion methods. For example, CONCAT fusion model outperforms attention-based fusion models in almost all the evaluated settings. This performance degradation is likely because nonverbal cues are interpreted independently from different perspectives. Moreover, as attention-based fusion approaches try to align multiview representations, the conflicting nonverbal cues from different perspectives lead to sub-optimal representations in these baseline models. We can develop models that can jointly consider multiple perspectives in extracting complementary representations from multiview data to address this issue.

**Results from human-subject study:** We sampled 300 data samples of the exo view from the testing split of CAESAR-XL to conduct a human-subject study on Amazon Mechanical Turk under IRB Protocol: 4627. In this study, we showed the data samples to participants and asked them to indicate whether a virtual avatar was pointing, gazing, and verbally describing the same object. Each sample was shown to three participants, and we took the majority voting to determine the label of a sample. In this study, 397 participants took part where each participant's task approval ratting was at least 95%, and they were compensated for their time. The results suggest that the participants correctly validated the relations in 80.66% of the times. Please check the supplementary document for details.

## 7 Broader Impact

We have developed an easy-to-use simulator for researchers to generate datasets for different purposes. We believe that datasets generated using our simulator can also be used to train and evaluate models for various tasks in embodied settings, such as embodied question answering, object grounding, and conversational human-AI interactions. Moreover, researchers can use Unity plugins to generate other modalities (e.g., a depth map, point clouds, and object segmentation) and annotations (e.g., 3D object spatial locations/rotations) for developing novel multimodal learning models. We expect researchers to be able to generate datasets according to their needs - which CAESAR's configurable parameters (specified in the supplementary) allow for. Additionally, CAESAR-generated datasets can be used to pre-train models for embodied instruction comprehension, which can be transferred to robots for comprehending instructions in real-world human-robot interactions. Finally, the findings from our experimental results open some exciting research directions to develop robust models for embodied referring expression comprehension.

## 8 Limitations and Future Works

Although we developed a 3D embodied environment in our simulator, we have rendered 2D image data in this work. In our future work, we will extend our simulator to render 3D data, such as point clouds. Using this 3D data we can develop models for multimodal instruction understanding in 3D embodied environments. Moreover, our experimental results from baseline methods suggest room for improvements in embodied referring expression understanding by using multimodal and multiview data. In the future, we plan to develop a model to extract complementary representations from multiple views and extract salient representations for nonverbal interactions to recognize embodied spatial relations accurately. Although we have developed and evaluated several visual-language transformer models, in future works, it will be an interesting avenue to investigate whether other visual-language models can effectively comprehend the embodied referring expressions. Specifically, as visual-language models take visual and verbal data as input together, it will be worth investigating whether these models can disentangle the nonverbal cues from the visual scene data and fuse the verbal data to produce salient multimodal representations.

## 9 Conclusion

In this work, we have introduced a novel embodied simulator, CAESAR, to generate referring expressions with verbal utterances and nonverbal cues. Our simulator captures nonverbal interactions from multiple views and generates verbal expressions from multiple perspectives (actor and observer). Using CAESAR, we have developed two large-scale datasets of embodied referring expressions. Our experimental results suggest that nonverbal cues improve model performance and that existing models cannot effectively learn multiple perspective-taking to ground embodied spatial relations accurately. We believe that our simulator will be helpful to generate situated interactional datasets and train models for diverse tasks in embodied settings.

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
