# OpenReview forum: "CAESAR: An Embodied Simulator for Generating Multimodal Referring Expression Datasets"
_NeurIPS.cc/2022/Track/Datasets_and_Benchmarks — NeurIPS 2022 Datasets and Benchmarks _

### Official Review · Reviewer_EnpK · 2022-07-03
**A dataset contribution where the task is to verify agreement between verbal referring expressions and nonverbal cues.**

**Rating:** 5
**Confidence:** 4
**Correctness:** The data construction, evaluation and…
**Clarity:** Yes

**Strengths:**

This paper correctly recognizes that,
- the perspective-taking ability, and
- the ability to infer nonverbal cues from multimodal input,

are the crucial missing components for adapting vision models, which were trained on non-embodied, static datasets, to an embodied environment.

This paper incorporates the above intuitions into their task design:
- **Nonverbal cues from multimodal input**: A model is required to compare the referred objects by nonverbal cues vs. the objects referred by verbal expressions.
- **Perspective-taking**: A model is required to perform the referring expression task while being viewpoint-aware (ego/exo/top), because spatial cues (e.g. to the left/right of) are inherently viewpoint-dependent.

This paper develops a simulator for data generation, as well as two generated datasets to facilitate research for more capable embodied agents.

**Weaknesses:**

1. CAESAR-L vs. CAESAR-XL, more explanation behind keeping two separate datasets would be great.
- What are the differences between -L and -XL except for the size-related factors (i.e. #samples, image resolution and image-to-sample ratio)?
- Do you expect these two datasets to support different evaluation scenarios or target different skillsets?
- Do you expect these two datasets to exhibit separate difficulty levels?
- Do you provide train/val/test splits? If yes, how do you make the split?
- If there is no significant difference, why not merging them into a single dataset?

2. Line272 "*force models to utilize nonverbal cues ... by recognizing which perspective did the given utterances come from*". I didn't get how the model is supposed to recognize that. When there are misalignments, one might consider two possibilities: 1) the current example is a "contrastive" example, where misalignment is anticipated, 2) it's necessary to fit oneself into another viewpoint. How do you expect a model to distinguish between these two possibilities? Please correct me if I'm misunderstanding. I'd appreciate the clarification.

3. (related to Q2) Line406 "*generates verbal expressions from multiple perspectives, actor and observer*". How could the model decide whether an input verbal expression comes from the actor or the observer? Do you expect the model to infer such information from the input images? And how?

4. Line302-307: I didn't get the essential difference between the Dual-encoder and the Late-fusion model besides using a different visual encoder (ViT vs. ResNet50). For example, does the Dual-encoder model perform any fusion earlier than the Late-fusion model?

5. Another clarification question: From where do you expect a model to extract gazing/pointing cues? Based on my current understanding, a model has to attend to the avatar's eyes and hands from the image input, correct? However, how could a gaze cue be extracted from a top or ego view? More explanation is greatly appreciated.

6. Line384 "*participants correctly validated the relations in 80.66% of the times*". It would be nice to provide some insight into when and where human participants failed at this task (accounting for ~20% of the examples). Could human failures be attributed to inherent ambiguity, unclear annotation instruction, the challenging nature of compositional reasoning, or any other reasons?

7. Table3: The best model performance is exceeding 80%, which is outperforming human participants. Where do you expect the future room for improvement to be?

**Additional Feedback:**

---

**Documentation:**

Detailed comparisons between the -L and the -XL versions will be appreciated, which will help potential users decide which version to work on.

**Ethics:**

No significant ethical concerns noticed.

**Relation To Prior Work:**

Yes

**Summary And Contributions:**

This work proposes 1) a new task for embodied referring expression, 2) a tool for easy-programming with a simulator for generating referring expression data, as well as 3) two generated datasets.

1) A new task for embodied referring expression.
- Two referring expressions are given in a verbal and a nonverbal (gazing/pointing/gazing+pointing) manner. The task is to decide whether or not the two referring expressions are indicating the same object.
- The nonverbal cues are provided to a model via rendered images of the scene from ego, exo, or top viewpoints. (**Please correct me if I'm wrong with regard to this point**)
- In order to perform this task, a model has to reason about spatial relationships described in the verbal cue, learn the links behind ego vs. exo vs. top viewpoints, and practice perspective-taking.

2) A tool for easy-programming with a simulator for data generation (CAESAR)
- CAESAR includes 80 objects and supports variations in color, size, position and rotation-angle.
- CAESAR is able to generate verbal referring expressions from 5 templates, which supports easy-to-difficult levels of compositional reasoning over object attributes and spatial relationships.
- CAESAR is able to synthesize the gesture and gaze of a humanoid avatar based on real human motions captured by OptiTrack

3) Two generated datasets
- A large (L) and an extra-large (XL) version
- The datasets can be used to test for understanding of nonverbal referring expressions as well as the perspective-taking ability of an embodied agent.

---

> ### Author Response · Authors · 2022-08-13
> **Response to Reviewer EnpK's comments (Part 1/4)**
>
> We thank the reviewer for the feedback. We appreciate that the reviewer agreed with us that perspective-taking and the ability to infer nonverbal cues from multimodal inputs as the crucial missing components in existing works.
> **Reviewer Comment:**
> ```
> CAESAR-L vs. CAESAR-XL, more explanation behind keeping two separate datasets would be great.
> What are the differences between -L and -XL except for the size-related factors (i.e. #samples, image resolution and image-to-sample ratio)?
> ```
> **Response:** As described in Section 4 of the manuscript, the primary differences between the CAESAR-L and CAESAR-XL are that in the CAESAR-L dataset human non-verbal interactions are rendered as RGB images, skeletal pose images, and videos whereas in the CAESAR-XL dataset human non-verbal interactions are only captured through RGB images. All these modalities are rendered for three views (ego, exo, and top) in both CAESAR-L and CAESAR-XL datasets.
>
> **Reviewer Comment:**
> ```
> -Do you expect these two datasets to support different evaluation scenarios or target different skillsets?
> -Do you expect these two datasets to exhibit separate difficulty levels?
> ```
> **Response:** Both datasets can be used for different tasks in embodied settings, such as object grounding, relation grounding, object segmentation, scene understanding, and visual question answering. There is no difference in evaluating these models trained on our datasets for the spatial relation grounding tasks. However, the model complexity can be different based on the input modalities. For example, if the models use video, then the model complexity will be increased compared to the models which use image modalities. Additionally, if researchers desire to extend the task, such as the keyframe identification task, only the CAESAR-L dataset would be able to be used for this purpose.
>
> **Reviewer Comment:**
> ```
> - Do you provide train/val/test splits? If yes, how do you make the split?
> - If there is no significant difference, why not merging them into a single dataset?
> ```
> **Response:** Although we divided these datasets into train, validation, and test splits and provided those splits CSV files with our datasets, we missed stating that information in the paper. We provided these splits as three CSV files (train.csv, valid.csv, and test.csv) inside the zip files of these datasets (CAESAR-L.zip and CAESAR-XL.zip). The first dataset, CAESAR-L, consists of 124,412 data samples created from 11,617,626 images at a resolution of 480x320 pixels. These data samples are divided into train, validation, and test data splits with 74760, 24779, and 24873 data samples, respectively. The second dataset, CAESAR-XL, consists of 1,367,305 data samples, which were created from 841,620 images by varying verbal expressions in the five different settings. These data samples are divided into train, validation, and test data splits with 1123886, 122157, and 121262 data samples, respectively. We have updated the manuscript, including these spits information (Section 4).
>
> Due to the large differences described it would not make sense or be possible to merge these datasets, as samples differ in contained information.
>
> We also provided data parsing scripts to parse the data and produce these splits (CAESAR-Source-Code.zip: data_parser-L.ipynb and data_parser-XL.ipynb). Please check the dataset zip files and data parsing scripts. Here are the screenshots and download links of these zip files. Additionally, following the suggestions of reviewer aFSL, we have developed another small dataset CAESAR-S. The data samples in CAESAR-S are randomly drawn from CAESAR-XL. Here is the summary of the splits of these datasets.
>
> | Datasets  | Total Images | Total Data Samples | Train Data Samples | Valid Data Samples | Test Data Samples |
> |---------------|--------------|--------------------|--------------------|--------------------|-------------------|
> | CAESAR-L  | 11,617,626   | 124,412            | 74,760             | 24,779             | 24,873            |
> | CAESAR-XL | 841,620      | 1,367,305          | 1,123,886          | 122,157            | 121,262           |
> | CAESAR-S  | 16,263       | 15,015             | 9,745              | 3,278              | 3,240             |
>
> The dataset can be downloaded using the following links:
>
> * CAESAR-XL dataset (319 GB): https://drive.google.com/file/d/13KAUBxW3jdu3RuUQMuJCamdNMa59gyXv
>
> * CAESAR-L dataset (181 GB): https://drive.google.com/file/d/1Q_kybqktCjthuIuWU01-l15muby69izq
>
> * CAESAR-S dataset (5.31 GB):
> https://drive.google.com/file/d/1eppJrUfxLNPQsY8s1ItsSlrYPRgJ6T0l/
>
> Source code of data processing, benchmark learning models: https://drive.google.com/drive/folders/1HRZrYgxDNi1wv9s0hNFBPJt51FACHgss
>
> Screenshots:
> https://drive.google.com/file/d/1xdqsmHtI2W1DAf63X8NF00M3x__CLQOm/view
> https://drive.google.com/file/d/1e8u-9cf-aAzfhFBFc1eBdnt7VLsvbDdp/view
> https://drive.google.com/file/d/1hDSVGjsd9-Pk5-053gsXxGJgkbFxlY2D/view

---

> > ### Comment · Reviewer_EnpK · 2022-08-25
> > **Good response!**
> >
> > I didn't realize that skeleton pose and video are only available in the -L version. That being said, it makes sense to keep separate versions. I appreciate that you created the -S version!

---

> ### Author Response · Authors · 2022-08-13
> **Response to Reviewer EnpK's comments (Part 2/4)**
>
> **Reviewer Comment:**
> ```
> Line272 "force models to utilize nonverbal cues ... by recognizing which perspective did the given utterances come from". I didn't get how the model is supposed to recognize that. When there are misalignments, one might consider two possibilities: 1) the current example is a "contrastive" example, where misalignment is anticipated, 2) it's necessary to fit oneself into another viewpoint. How do you expect a model to distinguish between these two possibilities? Please correct me if I'm misunderstanding. I'd appreciate the clarification.
> ```
>
> **Response:** These two possibilities are related. A verbal utterance can be interpreted from a speaker (ego) or observer (exo) perspective. The model needs to determine whether the referring expression is valid either from the speaker or observer from point of view. If nonverbal and verbal expressions are not coherent then the relationship is contrastive. As described in the paper (Section I), let us assume an actor is requesting an observer verbally to “pick up the left apple” (Fig. 1). This verbal expression can be interpreted differently from different perspectives, where the “left apple” from the exo view can be interpreted as the “right apple” from the ego view. In these scenarios, learning where the actor is looking at or pointing to can help identify whether the given verbal utterance holds the embodied relation from the speaker or observer's perspective (i.e., whether the person is verbally and nonverbally referring to the same object).
>
> **Reviewer Comment:**
> ```
> (related to Q2) Line406 "generates verbal expressions from multiple perspectives, actor and observer". How could the model decide whether an input verbal expression comes from the actor or the observer? Do you expect the model to infer such information from the input images? And how?
> ```
>
> **Response:** In these situations, the model needs to use both verbal utterances and nonverbal gestures to determine perspective and ground embodied relations. The model cannot effectively ground embodied relations without using nonverbal gestures because the interpretation of the verbal utterance depends on the perspective. For example, in the scenario mentioned in the previous response, learning where the actor is looking at or pointing to (information contained within the nonverbal modality) can help to identify the appropriate object. Thus, a model trained on these data samples should use both verbal utterances and non-verbal gestures to comprehend the referring expressions. Since existing datasets capture human interactions from only an exocentric perspective, models trained on these datasets can learn spurious correlations by exploiting perspective bias.
>
> **Reviewer Comment:**
> ```
> Line302-307: I didn't get the essential difference between the Dual-encoder and the Late-fusion model besides using a different visual encoder (ViT vs. ResNet50). For example, does the Dual-encoder model perform any fusion earlier than the Late-fusion model?
> ```
>
> **Response:** Dual-Encoder and Late-Fusion models have similar architectures as both of these models use separate encoders for visual and verbal modalities. However, there are two main differences between these models. First, Dual-Encoder models use Transformers to design both visual and verbal encoders. Contrastively, the Late-Fusion model uses different architectures for these two types of modalities, such as ResNet-50 for the visual encoder and BERT for the verbal encoder. Second, in the Late-Fusion model, we first extract visual representations for all three views and fuse the visual and verbal representations using a Transformer-style multi-head attention approach. For Dual-Encoder, we pair the verbal utterance to each view and pass each visual-verbal pair through the model to extract pairwise representations, which are concatenated to produce multimodal representations. We have updated the model architecture in the paper to clarify the difference between these two models (Section 5).

---

> > ### Comment · Reviewer_EnpK · 2022-08-25
> > **Model architectures for dual- vs. late-fusion are clarified! For the nonverbal cue question, can you confirm that my understanding below is correct?**
> >
> > Ideally, when the model is expected to figure out which one among ego/exo/top is correct:
> >
> > - "Top" is easier to recognize, while 'ego' vs. 'exo' is harder to distinguish.
> > - One possible way is to look at the arm positions.
> > - If the arm position is not visible, then a model might switch its belief between 'ego' and 'exo' once it observes something like a "flipped left/right descriptions".
> > - If neither 'ego' nor 'exo' seems right, then the sample should be considered a negative example.

---

> > > ### Author Response · Authors · 2022-08-27
> > > **Response to reviewer comment**
> > >
> > > The reviewer's suggested reasoning process can be one of the potential approaches for the learning model to validate the embodied relation. As the whole environment and non-verbal interactions are visible from the top view, it is easier for the model to recognize whether the embodied relation is valid from the top view (i.e., whether the human is verbally and non-verbally referring to the same object). Validating the embodied relation is challenging from exo and ego view, as the environment is not fully visible from these views. For example, the environment is partially visible from the ego view and the objects may be occluded from the exo view. Moreover, the arm is partially visible from the ego view.
> > >
> > > The results in Table 3 also support the similar phenomena that our evaluated models achieved improved performance in validating the embodied relations when the models use exo view compared to the ego view perspectives. Moreover, the models' performance improved using the top view perspective compared to the model using only exo view. However, the models' performance degrade when we provide multiple views compared to the model using only the top view perspective. The reason behind this performance degradation is that these models with multiple views as input need to validate the embodied relation from multiple perspectives, which requires learning additional reasoning from multiple perspectives. Thus, to ensure robust performance over multiple view perspectives, the models need to learn perspective in embodied settings effectively. These findings open future research directions in designing a learning model for comprehending embodied referring expressions from multiple verbal and visual perspectives.

---

> ### Author Response · Authors · 2022-08-13
> **Response to Reviewer EnpK's comments (Part 3/4)**
>
> **Reviewer Comment:**
> ```
> Another clarification question: From where do you expect a model to extract gazing/pointing cues? Based on my current understanding, a model has to attend to the avatar's eyes and hands from the image input, correct? However, how could a gaze cue be extracted from a top or ego view? More explanation is greatly appreciated.
> ```
>
> **Response:** Yes, if the model can attend to the human avatar’s eyes and hands, it can extract feature representations for gaze and pointing gestures to comprehend referring expressions effectively. However, as the avatar’s eyes are very small in captured RGB images, it is likely that the model will attend to the head angle to extract gaze cues much more easily than attending to the eyes. Thus, from the exocentric and top view perspectives, the model can use the human’s head angle to attend to gaze cues as well as the human’s arm/hand angles to attend to pointing gesture cues. From the egocentric perspective, the model can still observe the pointing gesture cue based on the arm/hand angle. However, since the angle of the human’s head is not directly visible from the egocentric view, the model must use the angle of the camera with respect to the table as well as the proximity of objects towards the center of the egocentric camera view to extract gaze cues. Consequently, we agree that it will be challenging to only use the ego view to determine the gaze as the model cannot directly see the head angle. That was one of the reasons we used multiple views so that the model learns to extract complementary multimodal representations to holistically understand embodied referring expressions.
>
> **Reviewer Comment:**
> ```
> Line384 "participants correctly validated the relations in 80.66% of the times". It would be nice to provide some insight into when and where human participants failed at this task (accounting for ~20% of the examples). Could human failures be attributed to inherent ambiguity, unclear annotation instruction, the challenging nature of compositional reasoning, or any other reasons?
>
> Table3: The best model performance is exceeding 80%, which is outperforming human participants. Where do you expect the future room for improvement to be?
> ```
> **Response:** We conducted the human study three times with different participants. We take the majority vote of three participants on a given question to determine the human participants' performance. As suggested by the reviewer, we went through the human study response and observed that at least one human participant correctly identified the embodied referring relation in 99.33% of cases. Moreover, as we go through the failed samples, it appears that the failed cases may be contributed by unfamiliarity with some of the uncommon objects, such as decahedrons and game consoles. We are not certain about this conclusion as we did not explicitly ask the human participants what makes it difficult to comprehend the embodied referring expressions. In future works, it will be interesting to investigate why the human participants could not accurately identify embodied relations for the failed samples. We plan to include an additional study where we show some samples and ask relevant questions to identify what makes it difficult for humans to understand embodied referring expressions. This study response may help to improve our simulator further. In general, the novelty and nonfamiliarity of some objects in the dataset to general participants with the task we created may have caused some confusion which may lead to part of the 20% error.
>
> The models outperformed the human performance for relation grounding tasks in some instances, such as when the models used the top view or when the models' used only gestures or gaze cues with verbal utterances. If we change the view from top to ego or exo the performance degrades considerably; in some cases, the accuracy drops to 60%. Thus, in order to ground embodied relations from multiple views, we must carefully design the model architecture and training procedure to extract cues from nonverbal modalities and effectively fuse these representations with verbal modalities to accurately recognize embodied spatial relations.

---

> > ### Comment · Reviewer_EnpK · 2022-08-25
> > **Good clarification! The authors' response reduced my concerns.**
> >
> > The author provided reasonable explanations for the 20% human error and more discussions on where models outperform humans. Though sometimes models are on par with humans, models are a lot less robust across views.
> >
> > Though it becomes a minor issue now, I still think it would make the work more solid to have detailed studies on humans' failure modes. We certainly want to rule out the possibility that the task is inherently ambiguous. But other failure modes like unfamilarity with certain objects are totally fine.

---

> > > ### Author Response · Authors · 2022-08-28
> > > **Future work with human's failure modes study**
> > >
> > > Thanks for understanding. As per reviewer's suggestions, we plan to conduct a detailed human study in our future work to analyze the impact of human's failure modes in comprehending embodied referring expressions.

---

> ### Author Response · Authors · 2022-08-13
> **Response to Reviewer EnpK's comments (Part 4/4)**
>
> **Reviewer Comment:**
> ```
> The nonverbal cues are provided to a model via rendered images of the scene from ego, exo, or top viewpoints. (Please correct me if I'm wrong with regard to this point).
> ```
>
> **Response:** Yes, models use rendered images (ego, exo, or top perspectives) capturing nonverbal interactions (gazes and pointing gestures) as well as verbal utterances (presented via text) to ground embodied relations. However, in the CAESAR-L dataset, we also generated skeletal pose images, which the model can use in addition to RGB images to extract the representations for nonverbal interactions. In the future, it will be an interesting avenue to extract complementary representations from multiple views, verbal utterances, and the skeletal pose captures to comprehend embodied referring expressions.

---

> > ### Comment · Reviewer_EnpK · 2022-08-25
> > **Thanks! The response is super clear.**
> >
> > ---

---

### Official Review · Reviewer_XoLN · 2022-07-04
**Review XoLN**

**Rating:** 6
**Confidence:** 3

**Strengths:**

I believe that the authors accurately identified the state of current datasets and benchmarks for referring expressions. Specifically, I think it is a good assessment to include a perspective as an important factor in the description of the given scene (which is the main differentiating factor with respect to YouRefIt [1]). I find that introducing a customisable tool that can generate any arbitrary number of samples for the task is valuable for the community. I also appreciate the inclusion of contrastive and ambiguous samples.

[1] Yixin Chen, Qing Li, Deqian Kong, Yik Lun Kei, Song-Chun Zhu, Tao Gao, Yixin Zhu, and Siyuan Huang. Yourefit: Embodied reference understanding with language and gesture. In Proceedings of the IEEE/CVF International Conference on Computer Vision, 2021

**Weaknesses:**

I find several aspects of the work that could be polished further, or provided with some more insight.

1. From my experience in Unity, I would recommend releasing the simulator as a standalone application. With new versions of Unity, packages sometimes become outdated w.r.t. the version of Unity and compatibility may be lost. I would suggest to the authors to consider that as a possibility to increase the sustainability of the simulator. Even though Unity offers to update packages when opened in a new version, it does not always resolve all updates correctly.
2. I would expect a bit more thorough description of the dataset samples: what is the number of various scenes, how many verbal cues are generated for each scene, what is the number of images generated for each scene/expression, what is the range on the number of generated frames when including video.
3. Further, the creation of contrastive samples could be slightly adjusted. Specifically, with respect to situations when the verbal cue is pointing to a non-existing object (line 227). I am not certain if that is very helpful in training the model. My doubt is what would be the expected output of the model. If making an inaccurate verbal description, I would consider miscueing only one property of the object (e.g. given only one mug, yellow, on the table, referring to it as 'the red mug') such that there is still information to be learned in the expression.
4. For the benchmark, if I understand correctly, a single verbal expression (from exo or ego perspective) is paired with all the views. Without providing any cue to which perspective verbal utterance corresponds, I think there may exist a lot of ambiguity in the samples, affecting the results.
5. On a smaller note, some comparison of using contrastive samples with the previous works (like [2]) could be useful. Also, some insights behind the choice of a specific engine from the visual quality perspective would be nice.

Finally, I find some things worth to be clarified (not necessarily weaknesses yet) - see Questions.

[2] Ankit Goyal, Kaiyu Yang, Dawei Yang, and Jia Deng. Rel3d: A minimally contrastive benchmark for grounding spatial relations in 3d. Advances in Neural Information Processing Systems, 2020

**Additional Feedback:**

Questions:
- In line 54 you refer to the typical human mistake of pointing to another object. Do you consider any constraints on when mistakes are made in contrastive examples? Intuitively, one would make a mistake probably when not using the name of the object (pointing to 'the mug' vs pointing to 'that red thing'), and the proximity of other objects sharing some visual property with the target could also induce mistakes (pointing to the incorrect object in the group of e.g. yellow objects).
- Another big source of confusion occurs when ego and exo views are opposite to each other and the actor tries to adjust the instruction to the observer's perspective ('my left or your left' scenario). Do you think it would be useful to consider such examples, or based on your experiments it would introduce too much noise into the system?
- What is the source of the assets in the scenes? Were objects modelled by the authors, or obtained from Unity Asset Store? If later, or other, is the use of the objects licensed? If the models were not created, do they overlap with any existing dataset?
- Could you explain synthesising the pointing gestures more? How many various 'pointing gestures' were collected by OptiTrack? When synthesising the gestures, do you reconstruct the joints as close to the collected ones as possible, or is there some noise added to slightly change the gesture? Finally, can the target of the referring expression exist only in one of the places corresponding to the set collected with OptiTrack?
- Are the orientations of objects saved in annotations? It would be useful in order to allow for full scene reconstruction from annotation only.
- For templates in Table 2, is it always ensured that the target is unique (especially for templates 1, and 2)?
- For Table 3b, do you think that performance of *Ego* may have been affected by the view not being able to utilise gaze cue, especially in the presence of various contrastive examples?

**Clarity:**

Overall, the paper is written fairly clearly, however, there are some sections that would require some more insight or disambiguation (see Questions).

**Correctness:**

The dataset is constructed in a sound way. The authors took care to balance the dataset with respect to the scenarios (templates) they propose, and also with respect to contrastive examples.


**Documentation:**

I believe the documentation is sufficient for use and reproducibility, the dataset is prepared and available for reviewers, and a docker environment eases the reproducibility. The only concern is what was raised in weaknesses - compatibility with future Unity versions.

**Ethics:**

I believe there are no ethical concerns.

**Relation To Prior Work:**

Relation to prior work is well covered, especially with respect to referring expressions datasets. Some more clarification with respect to various dataset generation tools (starting line 123) would be useful. Specifically, what prevents the transferability to other environments (line 128).

**Summary And Contributions:**

The authors of the paper propose a new embodied simulator developed in the Unity engine. The task of the simulator is to generate training data for multimodal referring expression task, i.e. containing verbal (language) and nonverbal (gesture, gaze) cues with respect to the target object. The authors point out the ambiguity of referring to objects in the real world - perspective, and mistakes in gestures, and address those while creating two proposed datasets with the use of their simulator. The authors provide various models and ablations explaining the shortcomings of current approaches to multimodal referring expressions.

---

> ### Author Response · Authors · 2022-08-13
> **Response to Reviewer XoLN's comments (Part 1/4)**
>
> We thank the reviewer for the valuable feedback. We are encouraged that the reviewer found our customizable simulator valuable for the research community. Moreover, we are happy that the reviewer appreciated the inclusion of contrastive and ambiguous samples in our generated datasets.
>
> **Reviewer Comment:**
> ```
> From my experience in Unity, I would recommend releasing the simulator as a standalone application. With new versions of Unity, packages sometimes become outdated w.r.t. the version of Unity and compatibility may be lost. I would suggest to the authors to consider that as a possibility to increase the sustainability of the simulator. Even though Unity offers to update packages when opened in a new version, it does not always resolve all updates correctly.
> ```
>
> **Response:** We thank the reviewer for this suggestion. We agree with the reviewer that releasing a standalone Unity application would be beneficial and likely prevent compatibility issues. We researched this solution ourselves as we developed the simulator and unfortunately, figured that such a solution is not feasible in our case. As we collect data in the CAESAR simulator using the Unity recorder, the editor version of Unity is required to record video, and thus, to generate data. Consequentially, we cannot export the Unity project to a standalone application. However, we have researched whether or not it would be possible to create a docker container for the Unity project. We believe this would be an easy way to reproduce the Unity environment and a sustainable way to ensure there are no future compatibility issues in our simulator. We plan to release a docker container in the future.
>
>
> **Reviewer Comment:**
> ```
> I would expect a bit more thorough description of the dataset samples: what is the number of various scenes, how many verbal cues are generated for each scene, what is the number of images generated for each scene/expression, what is the range on the number of generated frames when including video.
> ```
>
> **Response:** Thanks for the suggestions. In total, there are 11,617,626 scenes in the CAESAR-XL dataset and 841,620 scenes in the CAESAR-L dataset. We sample three verbal templates from 13 templates for each scene to generate verbal utterances. In total, each scene in the CAESAR-XL dataset (as this dataset only contains RGB images and no video) contains 15 images, with 3 images of each 4 nonverbal interactions settings (human using gaze and gestures, human using only gaze, human using only gestures, human using wrong gaze and gestures) as well as 3 images with no human settings. The number of images in the CAESAR-L dataset varies due to variable length videos, but the number of still canonical frames is 30 due to the inclusion of a skeletal pose modality. The number of generated frames for the CAESAR-L dataset (which was recorded at 15 FPS), ranges from 65 to 102. We have added this information to the supplementary document in Section 6.
>
>
> **Reviewer Comment:**
> ```
> Further, the creation of contrastive samples could be slightly adjusted. Specifically, with respect to situations when the verbal cue is pointing to a non-existing object (line 227). I am not certain if that is very helpful in training the model. My doubt is what would be the expected output of the model. If making an inaccurate verbal description, I would consider miscueing only one property of the object (e.g. given only one mug, yellow, on the table, referring to it as 'the red mug') such that there is still information to be learned in the expression.
> ```
>
> **Response:** We agree with the reviewer's suggestion that miscueing object property will be beneficial to training a robust model. We also thought in the same way to design the contrastive data generation system. We already used the reviewer’s suggested approach to generate contrastive data. We generated referring expressions for four settings: human using gaze and gestures; human using only gaze, human using only gesture; and no human in the scene. As we described in Section 3.4 in the paper, we generated the contrastive samples in two ways for these four settings. In the first approach, the human verbal and nonverbally (gaze and pointing gestures) referring two different objects present in the scene. These two objects can be the same object with different attributes (e.g., color, size, location). In the second approach, the verbally referred object is not present in the scene. We used the second approach only for the no-human avatar setting. The model trained on the data generated by the second approach should learn whether the object described verbally is present in the scene. These data samples can not only help to ground embodied relations but also help to learn object grounding in the scene.

---

> ### Author Response · Authors · 2022-08-13
> **Response to Reviewer XoLN's comments (Part 2/4)**
>
> **Reviewer Comment:**
> ```
> For the benchmark, if I understand correctly, a single verbal expression (from exo or ego perspective) is paired with all the views. Without providing any cue to which perspective verbal utterance corresponds, I think there may exist a lot of ambiguity in the samples, affecting the results.
>
> Another big source of confusion occurs when ego and exo views are opposite to each other, and the actor tries to adjust the instruction to the observer's perspective ('my left or your left' scenario). Do you think it would be useful to consider such examples, or based on your experiments it would introduce too much noise into the system?
> ```
>
> **Response:** We agree that a single verbal utterance paring with all the views introduce ambiguity. We intentionally introduce this ambiguity so that the model can learn the perspective to comprehend the referring expression using both verbal utterance and nonverbal gestures. These ambiguities introduced by generating verbal utterances from different perspectives can be resolved using non-verbal gestures. Let us assume an actor is requesting an observer verbally to “pick up the left apple” (Fig. 1). This verbal expression can be interpreted differently from different perspectives, where the “left apple” from the exo view can be interpreted as the “right apple” from the ego view. In these scenarios, learning where the actor is looking at or pointing can help identify the appropriate object. Thus, the model trained on these data samples should use both verbal utterances and non-verbal gestures to comprehend the referring expressions. On the other hand, as the existing datasets capture the human interactions from only an exocentric perspective, the models trained on these datasets can learn spurious correlations by utilizing the perspective bias.
>
>
> **Reviewer Comment:**
> ```
> On a smaller note, some comparison of using contrastive samples with the previous works (like [2]) could be useful. Also, some insights behind the choice of a specific engine from the visual quality perspective would be nice.
> ```
>
> **Response:** As we described in the related works section, in Rel3D [2] dataset, the generated visual scene contains only two objects for the spatial relation recognition task. Moreover, the Rel3D dataset does not include human nonverbal gestures and uses only verbal utterances to describe the relationship between objects. To generate the contrastive data samples, the Rel3D dataset varies object orientation. For example, a visual scene contains a clock and TV facing in opposite directions, whereas the verbal utterance is “clock is facing to TV.” On the other hand, our CAESAR simulator generates contrastive samples, where the verbal utterances and nonverbal gestures point to two different objects. Moreover,  CAESAR generates scenes with 4-10 objects to generate diverse data for referring expression comprehension.
>
> One of the primary reasons we chose the Unity engine was the High Definition Render Pipeline (HDRP). As this is the Render pipeline used in the CAESAR simulator, the HDRP offers very high-quality visuals and lighting as well as effects that we have used for extensions to our simulator, such as a depth map. The HDRP is comparable to the blender and unreal engine in terms of high-quality photo-realistic images.
>
> [2] Ankit Goyal, Kaiyu Yang, Dawei Yang, and Jia Deng. Rel3d: A minimally contrastive benchmark for grounding spatial relations in 3d. Advances in Neural Information Processing Systems, 2020
>
> **Reviewer Comment:**
> ```
> In line 54 you refer to the typical human mistake of pointing to another object. Do you consider any constraints on when mistakes are made in contrastive examples? Intuitively, one would make a mistake probably when not using the name of the object (pointing to 'the mug' vs pointing to 'that red thing'), and the proximity of other objects sharing some visual property with the target could also induce mistakes (pointing to the incorrect object in the group of e.g. yellow objects).
> ```
>
> **Response:** We consider several constraints for given contrastive examples, and both datasets also include another annotation - ambiguous - to address the different situations similar to the situations described. These constraints include a chosen object’s proximity to different copies of that chosen object, and the contrastive object selected being sufficiently spaced from the chosen object and of a different category. Objects are labeled as “ambiguous” when it is not clear which object the verbal and nonverbal referring expressions refer to. Ambiguous samples are explained in more detail in supplementary document Section 3.5, and the constraints regarding the generation of contrastive samples have been added to the paper in Section 3.4.

---

> ### Author Response · Authors · 2022-08-13
> **Response to Reviewer XoLN's comments (Part 3/4)**
>
> **Reviewer Comment:**
> ```
> What is the source of the assets in the scenes? Were objects modelled by the authors, or obtained from Unity Asset Store? If later, or other, is the use of the objects licensed? If the models were not created, do they overlap with any existing dataset?
> ```
>
> **Response:** Human models were downloaded from Mixamo, which are free to use. Moreover, Mixamo has other free assets that researchers can use to generate data. Objects in the scenes were downloaded from the Unity Asset Store, via four purchased packages linked below. As these object packages cost money, if other researchers would like to use these object libraries and extend those, they require to purchase those packages. However, if researchers wish to use other object libraries, paid or free, they may do so as well. The use of the object libraries we purchased is covered under a Single Entity license, whereas the Mixamo avatars we downloaded have no license. Although we select the object category by following the existing datasets, such as CLEVR, Rel3d, and YouRefIt, to our knowledge, the object models do not overlap with any existing datasets. We have added this information to the supplementary document in Section 3.3.
>
>
> * Mixamo: https://www.mixamo.com/
>
> * Cabin Interior Household Items and Furniture Pack With Interactive Component:
> https://assetstore.unity.com/packages/3d/props/furniture/cabin-interior-household-items-furniture-pack-with-interactive-c-138278
>
> * PBR Fruits and Vegetables HDRP:
> https://assetstore.unity.com/packages/3d/props/food/pbr-fruits-and-vegetables-hdrp-158808
>
> * Kitchen Accessories:
> https://assetstore.unity.com/packages/3d/props/interior/kitchen-accessories-200172
>
> * Supermarket Gluttony Pack:
> https://assetstore.unity.com/packages/3d/props/food/supermarket-gluttony-pack-12042
>
> **Reviewer Comment:**
> ```
> Could you explain synthesising the pointing gestures more? How many various 'pointing gestures' were collected by OptiTrack? When synthesising the gestures, do you reconstruct the joints as close to the collected ones as possible, or is there some noise added to slightly change the gesture? Finally, can the target of the referring expression exist only in one of the places corresponding to the set collected with OptiTrack?
> ```
>
> **Response:** Thirty various pointing gestures were collected using the OptiTrack. These synthesized gestures are referenced while creating new pointing gestures mathematically. Thus, we do add noise into different parameters to vary the gestures from the OptiTrack gesture data. No, the target of the referring expression can exist in any location, regardless of where the set collected with the OptiTrack was. The synthesis of pointing gestures is explained in detail in section 3.2 of the paper.

---

> ### Author Response · Authors · 2022-08-13
> **Response to Reviewer XoLN's comments (Part 4/4)**
>
> **Reviewer Comment:**
> ```
> Are the orientations of objects saved in annotations? It would be useful in order to allow for full scene reconstruction from annotation only.
> ```
>
> **Response:** No, the orientations of objects are not currently saved in annotations. However, we agree that including this data would be useful and now plan to add this feature onto future versions of the CAESAR simulator, especially since including such data would be extremely easy. Thank you for the insightful suggestion.
>
> **Reviewer Comment:**
> ```
> For templates in Table 2, is it always ensured that the target is unique (especially for templates 1, and 2)?
> ```
>
> **Response:** No, it is not always ensured that the target is unique for any of the templates. To ensure complete distinguishability of chosen objects inside of the CAESAR simulator, when generating objects and choosing a selected object we check that any objects of the same category of the target object are a sufficient distance away. This ensures that the generated nonverbal referring expression provides sufficient distinguishability between given scene objects, even with many copies of a given chosen object. Thus, if verbal referring expressions are insufficient to completely identify a target object the model is forced to use nonverbal referring expressions for the remainder of the task. We believe this feature allows for more realistic scenarios where there are multiple copies of a target object in a given scene while ensuring the model uses all modalities.
>
>
> **Reviewer Comment:**
> ```
> For Table 3b, do you think that performance of Ego may have been affected by the view not being able to utilise gaze cue, especially in the presence of various contrastive examples?
> ```
>
> **Response:** Yes, we agree that the performance was likely affected by the difficulty of utilizing the gaze cue. However, we would like to highlight that it is possible to utilize gaze cues from the ego view, as the view of the camera changes when the head moves during a gaze, which the model can observe with respect to the table. Additionally, the camera centers on the target object during a gaze as the head turns towards that object, so the model could also use this. However, due to the nature of the camera being non-static, it is likely that the model was unable to achieve this.
> Yeah, we think that not being able to utilize the gaze cues can degrade the models’ performance those are using only Ego view capturing human nonverbal interactions. Even our experimental analysis of the impact of modalities supports the effectiveness of using gaze cues. For example, incorporating only gaze cues improve the model's performance compared to those models that use verbal and visual modalities only (Table 3(a)).

---

### Official Review · Reviewer_1Jvt · 2022-07-18
**Large datasets and simulator that is novel and useful**

**Rating:** 8
**Confidence:** 4
**Correctness:** The dataset is constructed in a sound…
**Clarity:** The paper is clear and well written.

**Strengths:**

1. The paper presents two large scale datasets that have unique novelties compared to all existing datasets, such as these datasets provide ego (first-person), exo (third person) and top views of the scene where all previous datasets only provide exo views, and these two new datasets provide contrastive samples and ambiguous samples. These novel features can benefit research on interactive AI,
2. The authors not only provide the datasets, but also release a configurable simulator for generating the datasets, so if researchers are unsatisfied by the configurations used to generate the two provided datasets, they can use the simulator to generate their own data with their desired configurations.
3. The authors benchmarked their datasets on a few SOTA multimodal models such as CLIP, ViT+BERT, etc, and studied the impacts of modalities as well as the multiple perspectives, thus pointing out advantages of the datasets as well as providing potential future research directions for users of the datasets.
4. The authors carefully documented details of their dataset collection process and how to use their simulator tools in the supplemental materials.


**Weaknesses:**

N/A

**Additional Feedback:**

N/A

**Documentation:**

The authors documented sufficient details in data collection, organization, availability, maintenance, ethical and responsible use in the paper and the supplement material. There is very detailed description of every step in the dataset generation and curation process in the supplemental materials.

**Ethics:**

No ethical concerns.

**Relation To Prior Work:**

The paper very clearly discussed how it differs from previous contributions (which is very clearly summarized in Table 1)

**Summary And Contributions:**

The paper presents CAESAR, a simulator capable of automatically generating multimodal referring expressions with verbal utterances and nonverbal cue (pointing gesture and gaze) to refer to an object. The major contributions of the paper includes the CAESAR simulator, as well as two large-scale datasets CAESAR-L and CAESAR-XL, with over a million samples in total. The most important advantage of these datasets compared to existing datasets is that these datasets provide ego (first-person), exo (third person) and top views of the scene where all previous datasets only provide exo views. The authors benchmarked various models on the datasets.

---

> ### Author Response · Authors · 2022-08-13
> **Thanks for appreciating our work and contributions.**
>
> We thank the reviewer for the insightful feedback. We are encouraged that the reviewer found that our configurable simulator and novel large-scale datasets can benefit research on interactive AI. The reviewer also acknowledges that the datasets are constructed in a sound way, the paper is well written, and we provide sufficient details in data collection, organization, availability, maintenance in the paper, and the supplement materials.

---

### Official Review · Reviewer_Fxyr · 2022-07-20
**The authors developed a novel simulator for generating multimodal referring expressions and provided two large-scale datasets with the benchmarks. This work effectively solves the problem of the previous dataset always ignoring the perspective-awareness. This work provides excellent convenience and hints of research directions for subsequent investigators.**

**Rating:** 7
**Confidence:** 4
**Clarity:** The paper is well written with detail…

**Strengths:**

1. The data generation system developed in this work can help increase the model's robustness with the awareness of multiple perspectives.
2. The proposed data generation system is easy to use and user-friendly, and open to researchers.
3. A brief summary of the limitations and future works are given, which provides new inspiration for subsequent work in this area.

**Weaknesses:**

 In the experiment result shown in Table 3, adding the result of combining two different perspectives and analyzing the results will be more convincing for this part.

**Additional Feedback:**

Non.

**Correctness:**

The claims made in the submission correct and the datasets are constructed in a sound way.

**Documentation:**

The detail of the data collection and organization is sufficient, and there are no problem for the availability.

**Ethics:**

Non ethical concerns.

**Relation To Prior Work:**

This work summarizes the problems and shortcomings of previous work. It proposes data generation models based on these problems to provide data and ideas for subsequent training of more robust models.

**Summary And Contributions:**

1. The CAESAR developed by the authors break the limitations of the previous dataset for capturing embodied multimodal referring expressions by providing the awareness of multiple perspectives.
2. The data generation system is easy to use and can provide large convenience for other researchers.
3. The use of this data generation system for multi-perspective awareness is an enlightening breakthrough for current research in the field.
4. Benchmarks with SOTA methods are done with the provided datasets for grounding embodied spatial relations.

---

> ### Author Response · Authors · 2022-08-13
> **Response to Reviewer Fxyr's comments**
>
> Thanks for the feedback. We are encouraged that the reviewer finds our data generating system easy to use, user-friendly, and the generated data can help to increase the model’s robustness with the awareness of multiple perspectives.
>
> **Reviewer Comment:**
> ```
> In the experiment result shown in Table 3, adding the result of combining two different perspectives and analyzing the results will be more convincing for this part.
> ```
>
> **Response:** In our experimental analysis, we have evaluated the impact of combining three perspectives (ego, exo, and neutral) in the baseline models. We used the verbal utterances from ego, exo, and neutral perspectives to evaluate whether the baseline models can effectively learn perspective to ground embodied referring relation.
>
> The results in Table 3(b) suggest that the models' performance varies with the perspectives. For example, the top view improves the performance of Late Fusion and Dual Encoder models compared to models using the exo view. These findings underscore that the exo view is not always the optimal perspective for ensuring robust performance. Additionally, although the performance of the single view-based models fluctuates, incorporating multiple perspectives helps achieve consistent performance across the models. In our experiments, we found that multiview models generally cannot outperform single view-based models. The reasoning behind this performance degradation is that nonverbal cues can be interpreted differently from multiple views, which current models do not take into account. For example, a person pointing and verbally describing an object as the "left apple" can be visually interpreted by an observer as the "right apple". Thus, extracting synchronized cues across modalities is essential to validating an embodied spatial relation. Moreover, the evaluated baseline models do not explicitly learn to ground perspective to comprehend referring expressions. As the referring expressions in our datasets are generated from multiple perspectives, our datasets can be used to diagnose whether a model can effectively learn perspective-taking to comprehend embodied referring expressions.
>
> We have updated the manuscript to clarify the experimental settings and provide additional insights. Please check Section 6 in the updated manuscript.

---

> > ### Comment · Reviewer_Fxyr · 2022-08-29
> > **The authors' response dispelled my doubts.**
> >
> > Thank you for your responses. I think the revised version is more precise.

---

### Official Review · Reviewer_aFSL · 2022-07-22
**Elementary but crucial work toward human science pedagogy and recognition**

**Rating:** 7
**Confidence:** 4

**Strengths:**

I do appreciate the dataset synthesizing method: CAESAR bridges the simulation in a virtual engine with motion captures in reality, making it possible to run experiments on testing the embodied AI's performance in real indoor settings.

The data collecting process is clear but non-trivial. Compared with previous related work on collecting/synthesizing pointing and gazes (e.g. ReferIt3D, CLEVR-Ref), the CAESAR-L/XL dataset contains thorough descriptive information for the agent, as well as for the scene/object understanding.

As a dataset plus simulator, the dataset is well-organized. The website for holding the benchmark is well-designed.

**Weaknesses:**

The thing I worry about most about the paper is the generalizability of the dataset. (a) Even though the dataset is of huge size, it is based on few scene types. I am wondering if is easy for the dataset to be synthesized in photo-realistic indoor scenes. (b) For the 80 objects in the dataset, in my opinion, they mostly are small items in the kitchen or living room. How can we believe they are representative of embodied AI tasks. (c) The dataset brings human gestures and gazes, but I don't see this work describing the variability when humans are pointing and looking at an object.

For the experiments, this work runs several end-to-end experiments on grounding the referential task given inputs of multi-modalities. However, I want to know (a) What is the insight when we bring gaze, pointing, language, and views together? (b)How do pointing and gaze cooperate to bring or remove ambiguity from language descriptions?

**Additional Feedback:**

The dataset size is extremely large. Do you plan to release some sample data of a small size to help users to get started?

**Clarity:**

The paper is well written, except that the figures and tables are surrounded by texts (looks crowded).

**Correctness:**

The paper presents the main contribution as datasets. The process of collecting the data is demonstrated in detail. Statistics from exploratory data analysis are well presented in the paper and appendix. The experiments and evaluation methods are straightforward and clearly defined.

**Documentation:**

There are sufficient details on data collection, organization, and availability. The data can be shared via drive and getting-start guidance is provided via the video and slides. However, I really want to see better documentation containing the maintenance plan.

**Ethics:**

I don't see any concerns regarding ethics.

**Relation To Prior Work:**

The embodied AI setting (egocentric view) makes this work different from most previous visual understanding works (e.g. RefCoCo, Clever, YouRefIt). As far as my knowledge goes, the motion tracking + synthesizing data collecting strategy is original and unique in the related fields.

**Summary And Contributions:**

This work introduces a new embodied multi-model referring simulator (CAESAR) for understanding human verbal utterance, pointing, and gaze.
Two datasets of varying sizes have been collected from CAESAR containing resourceful information for referential tasks.
Besides, several baselines about configuring the grounding tasks are performed to illustrate the different impacts of different modalities.

---

> ### Author Response · Authors · 2022-08-13
> **Response to Reviewer aFSL's comments (Part 1/3)**
>
> We thank the reviewer for the valuable suggestions. We are happy that reviewers appreciate our dataset synthesizing method, which bridges the simulation in a virtual engine with data from a motion capture system for evaluating embodied AI systems’ performance in real indoor settings. Moreover, we are motivated that the reviewer finds our datasets well-organized, benchmarks are well-designed, and our simulator is non-trivial compared to the existing works.
>
> **Reviewer Comment:**
> ```
> The dataset size is extremely large. Do you plan to release some sample data of a small size to help users to get started?
> ```
>
> **Response:** Thanks for the suggestion. We have developed another small dataset CAESAR-S. The data samples in CAESAR-S are randomly drawn from CAESAR-XL. CAESAR-S consists of 16263 data samples created from 15015 images by varying verbal expressions in the five different settings. These data samples are divided into train, validation, and test data splits with 9745, 3278, and 3240 data samples, respectively. The dataset can be downloaded using the following link. We will release CAESAR-S to help users to get started.
>
> CAESAR-S dataset (5.31 GB):
> https://drive.google.com/file/d/1eppJrUfxLNPQsY8s1ItsSlrYPRgJ6T0l/
>
> **Reviewer Comment:**
> ```
> The thing I worry about most about the paper is the generalizability of the dataset.
> (a) Even though the dataset is of huge size, it is based on few scene types. I am wondering if is easy for the dataset to be synthesized in photo-realistic indoor scenes.
> ```
>
> **Response:** We varied the scenes regarding object category, object attributes (color, shape, and locations), and environment attributes (color and lighting conditions). Moreover, we have generated data samples from different perspectives (speaker and observer). Additionally, we captured the human interactions from multiple views (ego, exo, and top). These various environment settings generated diverse data samples to train a robust model. Moreover, as our generated data samples are not biased towards a particular perspective, the model trained on our datasets should need to learn perspective and should need to learn generalized representations to comprehend referring expressions accurately from multiple perspectives. On the other hand, the existing datasets captured the human interactions solely from an exocentric perspective, which can bias the trained model to a particular perspective and can not effectively comprehend the referring expression from multiple perspectives. Thus, our dataset can help to develop a robust model which can learn generalized multimodal representation to comprehend referring expressions in embodied settings.
>
> Additionally, we plan to extend our simulator to include more diverse environments, such as offices, stores, and libraries, to generate more diverse data samples. These data samples can help to ensure that the model should learn generalized representations to comprehend referring expressions in diverse embodied settings. We designed the simulator in an extensible and modular way so that new environments (e.g., store, library, office, etc.) can be easily incorporated into the simulator to design more diverse data samples. We are continually working on incorporating more environments into the simulator, such as the store environment. These new environments will allow for studying more diverse spatial relationships as well as ensure better model generalizability to real-world scenarios. In the same way, researchers can create photo-realistic indoor environments for our simulator and generate data with these new environments.
>
> **Reviewer Comment:**
> ```
> (b) For the 80 objects in the dataset, in my opinion, they mostly are small items in the kitchen or living room. How can we believe they are representative of embodied AI tasks.
> ```
>
> **Response:** We choose the object categories carefully to develop a diverse object library. Our object library contains objects we generally use in our daily life, such as kitchen and living room items, but also contains uncommon items, such as decahedrons and VR devices. Our simulator also allows the inclusion of additional objects in the object library to generate referring expressions. Additionally, as the home and kitchen items are diverse and graspable, the model trained on these data samples can be used in robot learning to implement human-robot interactive systems. We have updated the supplementary document to include the reasoning behind choosing these object categories.

---

> ### Author Response · Authors · 2022-08-13
> **Response to Reviewer aFSL's comments (Part 2/3)**
>
> **Reviewer Comment:**
> ```
> (c) The dataset brings human gestures and gazes, but I don't see this work describing the variability when humans are pointing and looking at an object.
> ```
>
> **Response:** As described in Section 3.2 of the manuscript (particularly Pointing Gesture Synthesis), one of the primary goals of the synthesis of pointing gestures in our simulator was to match the variability of real-world points. Thus, we used learned parameters from real human motion data to introduce variability for synthesizing pointing gestures to refer to an object in the scene. From these learned parameters, we apply inverse kinematics as well as noise to render gestures. This gesture synthesis approach allows CAESAR to generate diverse nonverbal interactions with high variability. We have updated the manuscript to clarify how we ensure variability while generating gestures [Section 3.2].
>
> **Reviewer Comment:**
> ```
> For the experiments, this work runs several end-to-end experiments on grounding the referential task given inputs of multi-modalities. However, I want to know (a) What is the insight when we bring gaze, pointing, language, and views together?
> ```
>
> **Response:** As we described in Section 6, the results in Table 3(a) suggest that a verbal model without any nonverbal signals (e.g., BERT) can not perform better than random guessing at the relation grounding task. The reasoning behind this performance is that, in CAESAR-L and XL, we generated contrastive nonverbal data samples using the same verbal utterance. The results also suggest that incorporating nonverbal modalities (gaze and pointing gesture) improved embodied spatial relation grounding accuracy. For example, incorporating gaze cues improve the model's performance compared to those models that use verbal modalities only. This performance improvement indicates the necessity of nonverbal modalities to ground embodied spatial relations. However, the performance degrades when models use both gaze and gesture nonverbal cues compared to models using only gaze or gesture. As the evaluated baseline models extract monolithic representations for pointing gestures and gaze, these models may not effectively disentangle pointing gesture and gaze representations to fuse to the verbal representations. Thus, in future work, we plan to investigate how to effectively extract cues from nonverbal modalities and fuse these representations with verbal modality to recognize embodied spatial relations accurately.
>
> We provided additional insights into the usage of gaze, pointing gestures, and verbal utterance modalities to comprehend referring expressions in the first part of Section 6.
>
>
> **Reviewer Comment:**
> ```
> (b) How do pointing and gaze cooperate to bring or remove ambiguity from language descriptions?
> ```
>
> **Response:** As we described in Section 1, the aim of including pointing gestures is to complement the verbal modality for accurately comprehending referring expressions. The comprehension of referring expressions using only verbal utterances varies with the perspective and is often ambiguous with solely verbal utterances. In ambiguous cases, the pointing gesture and gaze complement verbal utterances to accurately ground objects and relations in embodied settings. Let us assume that an actor is requesting an observer verbally to “pick up the left apple” (Fig. 1). This verbal expression can be interpreted differently from different perspectives, where the “left apple” from the exo view can be interpreted as the “right apple” from the ego view. In these scenarios, learning where the actor is looking at or pointing to can help identify the appropriate object. Additionally, there are cases where people use incomplete verbal expressions - e.g., the knife next to the cutting board, when there may be multiple knives and cutting boards, where the inclusion of a pointing gesture and gaze can clarify this ambiguity. Thus our generated data samples with multimodal cues (verbal utterance and nonverbal gestures) enable a model to learn perspective-taking to ensure seamless and natural interactions in embodied settings.

---

> ### Author Response · Authors · 2022-08-13
> **Response to Reviewer aFSL's comments (Part 3/3)**
>
> **Reviewer Comment:**
> ```
> I really want to see better documentation containing the maintenance plan.
> ```
>
> **Response:** We use GitHub to host and publicly release all the source code for the simulator, dataset parsing, and benchmark model experimentation. We will release the future versions of our simulator through GitHub as well. Moreover, we will encourage other researchers to create a pull request to update the simulator, issue bugs, and resolve any known bugs. The researchers can also request additional features through pull requests on GitHub.
>
> We host our datasets in two places: Google Drive and the University of Virginia’s secure storage system (Rivanna: https://www.rc.virginia.edu/userinfo/storage/). As the Rivanna storage system periodically backups the data, this storage is safer and ideal for long-term data storage. We will share the datasets with other researchers via the Google drive links or provide read-only access to the Rivanna storage system using the Globus File share system (https://www.globus.org/data-sharing). Additionally, as our generated datasets are large in size, if researchers prefer, we can share the dataset via storage drives. However, the researchers need to bear a fixed charge to cover the cost of the storage drive and standard shipping costs. The authors of the paper maintain both of the generated datasets. Contact person: Md Mofijul Islam (Email: mi8uu@virginia.edu) and Tariq Iqbal (tiqbal@virginia.edu).
>
> We have included a section in the supplementary document to describe the simulator and datasets maintenance plan in detail. (Supplementary Document Section 2).

---

### Official Review · Reviewer_K8km · 2022-07-28

**Rating:** 7
**Confidence:** 3
**Clarity:** Yes

**Strengths:**

1. Table 1 in the paper is extremely well designed, and the information is succinctly conveyed. Everything I need to know about this work in comparison to existing referring expression datasets is neatly summarized in this one table. The related work is also very well covered.

2. The methodology for designing the simulator is described in great detail, and very easy to follow. All the design decisions are motivated well. It is clear the authors put a great deal of thought into possible issues with the dataset.

3. The dataset analysis is very well done, and gives useful insights.

4. The authors release not only their generated datasets but also their simulator.

5. The paper is well-written, and very easy to follow.


**Weaknesses:**

1. This seems to be a glaring omission: for the two generated datasets, CAESAR-L and CAESAR-XL, no train/val/test splits are reported. Does that mean the models were trained and evaluated on the full data?? If so, I can't take any of the results in Section 6 at face value.

2. While it is great that the authors are releasing not only their generated datasets but also the simulator, it's not clear how the authors think this simulator can be useful to other researchers. What can the simulators be used to design that is not already covered by the generated CAESAR datasets? This is not a hard requirement, or even a real weakness, but would be useful to motivate releasing the simulator, or as Future Work discussion.

3. The paper could use more vision-language pre-trained model baselines, such as UNITER or ViLT, or any other VL transformer model that jointly takes vision and language inputs and processes them in the Transformer together.

**Additional Feedback:**

-

**Correctness:**

It appears to be constructed in a sound way - lack of train/val/test splits notwithstanding.

**Documentation:**

Yes

**Ethics:**

-

**Relation To Prior Work:**

Yes

**Summary And Contributions:**

The authors introduce CAESAR, an embodied simulator for generating multimodal referring expression datasets. The generated referring expressions can include both verbal utterances and nonverbal gestures (pointing + gaze), and require resolving spatial relations between objects in the scene. The simulator can also be used to generate contrastive examples, where the referred object from the verbal utterance does not match the object referenced by the nonverbal gesture. The authors use their simulator to generate two datasets, and benchmark various existing methods on these generated datasets.

I am mostly happy with this work, but I would like weakness #1 to be addressed since this seems like a glaring blindspot.

---

> ### Author Response · Authors · 2022-08-13
> **Response to Reviewer K8km's comments (Part 1/2)**
>
> We thank the reviewer for the valuable feedback. We are motivated that the reviewer is happy about our work. We are delighted that reviewers find our paper well designed, the development of different parts of the simulator are described in great detail, and datasets analysis provides useful insights. We are happy that reviewers appreciate our effort to develop our simulator and release this simulator publicly for other researchers.
>
> **Reviewer Comment:**
> ```
> This seems to be a glaring omission: for the two generated datasets, CAESAR-L and CAESAR-XL, no train/val/test splits are reported. Does that mean the models were trained and evaluated on the full data?? If so, I can't take any of the results in Section 6 at face value.
> ```
> **Response:**
> Thanks for pointing out this missing information. Although we divided these datasets into train, validation, and test splits and provided those splits CSV files with our datasets, we missed to explicitly state that information in the paper. We provided these splits as three CSV files (train.csv, valid.csv, and test.csv) inside the zip files of these datasets (CAESAR-L.zip and CAESAR-XL.zip).
>
> We have also provided data parsing scripts to parse the data and produce these splits (CAESAR-Source-Code.zip: data_parser-L.ipynb and data_parser-XL.ipynb). Please check the dataset zip files and data parsing scripts. Here are the screenshots and download links of these zip files. We have added these spits information in the paper (Section 4) and the supplementary document.
>
> Additionally, following the suggestions of reviewer aFSL, we have developed another small dataset CAESAR-S. The data samples in CAESAR-S are randomly drawn from CAESAR-XL.
>
>
> Here is the summary of the splits of these datasets.
>
> | Datasets  | Total Images | Total Data Samples | Train Data Samples | Valid Data Samples | Test Data Samples |
> |---------------|--------------|--------------------|--------------------|--------------------|-------------------|
> | CAESAR-L  | 11,617,626   | 124,412            | 74,760             | 24,779             | 24,873            |
> | CAESAR-XL | 841,620      | 1,367,305          | 1,123,886          | 122,157            | 121,262           |
> | CAESAR-S  | 16,263       | 15,015             | 9,745              | 3,278              | 3,240             |
>
> The dataset can be downloaded using the following links:
>
> * CAESAR-XL dataset (319 GB): https://drive.google.com/file/d/13KAUBxW3jdu3RuUQMuJCamdNMa59gyXv
>
> * CAESAR-L dataset (181 GB): https://drive.google.com/file/d/1Q_kybqktCjthuIuWU01-l15muby69izq
>
> * CAESAR-S dataset (5.31 GB):
> https://drive.google.com/file/d/1eppJrUfxLNPQsY8s1ItsSlrYPRgJ6T0l/
>
> Source code of data processing, benchmark learning models: https://drive.google.com/drive/folders/1HRZrYgxDNi1wv9s0hNFBPJt51FACHgss
>
> Screenshots:
> https://drive.google.com/file/d/1xdqsmHtI2W1DAf63X8NF00M3x__CLQOm/view?usp=sharing
> https://drive.google.com/file/d/1e8u-9cf-aAzfhFBFc1eBdnt7VLsvbDdp/view?usp=sharing
> https://drive.google.com/file/d/1hDSVGjsd9-Pk5-053gsXxGJgkbFxlY2D/view?usp=sharing
>
> **Reviewer Comment:**
> ```
> While it is great that the authors are releasing not only their generated datasets but also the simulator, it's not clear how the authors think this simulator can be useful to other researchers. What can the simulators be used to design that is not already covered by the generated CAESAR datasets? This is not a hard requirement, or even a real weakness, but would be useful to motivate releasing the simulator, or as Future Work discussion.
> ```
> **Response:**
> One of the primary goals of designing this simulator is to allow other researchers to extend this simulator to generate virtually any amount of data from diverse environments. For example, this simulator can be extended to generate embodied interactions in other environments, such as kitchens, homes, stores, offices, etc. Moreover, the CAESAR simulator can be extended to generate data samples and annotations for various tasks in embodied settings, such as embodied question answering, object grounding, and perspective grounding. Additionally, researchers can use Unity plugins to generate other modalities (e.g., depth and point cloud) and annotations (e.g., object segmentations) for developing novel multimodal learning models. Lastly, we wanted researchers to be able to generate datasets according to their desires - which in the case of CAESAR’s configurable parameters specified in Section 4 of the supplementary document, include the presence of video, generating different modalities of image representations of scenes, using different object libraries, and using different environmental parameters (i.e., floors, tables, humans). We have updated the manuscript to describe our simulator's impact and additional usage in detail (Section 7).

---

> ### Author Response · Authors · 2022-08-13
> **Response to Reviewer K8km's comments (Part 2/2)**
>
> **Reviewer Comment:**
> ```
> The paper could use more vision-language pre-trained model baselines, such as UNITER or ViLT, or any other VL transformer model that jointly takes vision and language inputs and processes them in the Transformer together.
> ```
>
> **Response:** Thanks for the suggestions about evaluating other visual-language transformer models. We have evaluated several visual-language models in our original evaluation, such as CLIP, ViT+BERT. The results are presented in Table 3. We have also evaluated several fusion methods, such as CONCAT, SUM, Self-Attention, and Cross-Attention. The existing visual-language transformer models predominantly use a combination of Self-Attention and Cross-Attention methods. Although we have developed and evaluated several visual-language transformer models, in future works, it will be an interesting avenue to investigate whether other visual-language models can effectively comprehend the embodied referring expressions. Specifically, as the visual-language models take visual and verbal data as input together, it will be worth investigating whether these models can disentangle the nonverbal cues from the visual scene data and fuse to the verbal data to produce salient multimodal representations. We have updated the paper (Section 8: Limitations and Future Works) to reflect these possible avenues of research areas where our simulator can be used to evaluate the visual-language model to comprehend embodied referring expressions.

---

> > ### Comment · Reviewer_K8km · 2022-08-18
> > **Response to Response**
> >
> > Hello, thank you for the response. As you mention, you evaluate both CLIP and ViT+BERT, and use several fusion methods to merge the representations coming out of each modality. However, these are dual-encoder VL models, where language and vision are processed separately. I was specifically asking about single-encoder models like UNITER and ViLT which process language and vision inputs jointly in the same Transformer.

---

> > > ### Author Response · Authors · 2022-08-22
> > > **Working on progress**
> > >
> > > Thanks for the suggestion. We are working to develop and evaluate visual-language transformer models on our datasets, particularly ViLT, UNITER, and VisualBERT. As soon as we can complete the model development, training, and evaluations, we will share the results with you. We are hopeful that the evaluations will be ready by the author-reviewer discussion deadline (August 29). However, as training the models on these datasets often require a long time, in the worst case, if we cannot complete the evaluation before then, we will share the final results of these models with the camera-ready paper.

---

> > > > ### Author Response · Authors · 2022-08-27
> > > > **Additional experimental evaluations using single visual-language encoders**
> > > >
> > > > Thanks again for the suggestion. We have extended two state-of-the-art visual-language transformer models, ViLT and VisualBERT, and conducted experimental analysis for grounding embodied relations on our dataset CAESAR-XL. We have updated the paper and included these models' architecture details and the experimental results. Please check Table 3 in the updated paper.

---

### Meta-Review · Area_Chair_gSUS · 2022-09-09

**Recommendation:** Accept
**Confidence:** 4

**Metareview:**

All the reviewers appreciated the effort and rigour that went into the design of the proposed idea. The presentation of the paper clearly shows the embedding of the work within the state of the art. It is particularly appreciated that this work is likely to lead to a break through in the analysis of nonverbal referring expressions given the possibility to generate multiple perspectives synthetically. The experiments compared with multiple state of the art models and modalities was also appreciated.

The answers by the authors to reviewer comments were thorough and provided additional insights.

Not all reviewers were in agreement on some aspects of the work and the more negative comments questioned whether releasing the simulator will really have the impact that it could have on first appearances. (XoLN)

enpk's final response seems more positive than the rating so I think that they may have forgotten to update their score.
The concerns of XoLN regarding the practical details of releasing a simulator are to be considered. However, it is not clear to me from the paper whether releasing the simulator is really considered a major contribution of the work rather than a minor contribution.

---

### Decision · Program_Chairs · 2022-09-16

Accept